# Insulin-producing organoids engineered from islet and amniotic epithelial cells to treat diabetes

Fanny Lebreton [1,7], Vanessa Lavallard [1,7], Kevin Bellofatto[1,7], Romain Bonnet[1], Charles H. Wassmer[1], Lisa Perez[1], Vakhtang Kalandadze[2,3], Antonia Follenzi [3], Michel Boulvain[4], Julie Kerr-Conte[5], David J. Goodman[6], Domenico Bosco [1], Thierry Berney [1] & Ekaterine Berishvili [1,2]*

Maintaining long-term euglycemia after intraportal islet transplantation is hampered by the considerable islet loss in the peri-transplant period attributed to inflammation, ischemia and poor angiogenesis. Here, we show that viable and functional islet organoids can be successfully generated from dissociated islet cells (ICs) and human amniotic epithelial cells (hAECs). Incorporation of hAECs into islet organoids markedly enhances engraftment, viability and graft function in a mouse type 1 diabetes model. Our results demonstrate that the integration of hAECs into islet cell organoids has great potential in the development of cell-based therapies for type 1 diabetes. Engineering of functional mini-organs using this strategy will allow the exploration of more favorable implantation sites, and can be expanded to unlimited (stem-cell-derived or xenogeneic) sources of insulin-producing cells.

[1] Cell Isolation and Transplantation Center, Surgery, Geneva University Hospitals and University of Geneva, Geneva, Switzerland. [2] Institute of Medical Research, Ilia State University, Tbilisi, Georgia. [3] Department of Health Sciences, University of Oriental Piedmont, Novara, Italy. [4] Department of Gynecology and Obstetrics, Geneva University Hospitals and University of Geneva, Geneva, Switzerland. [5] INSERM U1190, Translational Research for Diabetes, University of Lille, Lille, France. [6] Department of Nephrology, St Vincent's Hospital, Melbourne, VIC, Australia. [7] These authors contributed equally: Fanny Lebreton, Vanessa Lavallard, Kevin Bellofatto. *email: ekaterine.berishvili@unige.ch

Although intraportal islet transplantation is an established therapy for patients with type 1 diabetes, maintaining long-term glucose control with this approach remains challenging, mainly due to considerable islet loss in the peri-transplant period[1]. Among the reasons for early graft loss, inflammation at the site of implantation and impaired revascularization appear as key factors[2–4]. Pancreatic islets have a dense blood supply which is inevitably disrupted by the isolation process[5]. In the first weeks after transplantation, oxygen and nutrients are delivered to avascular islets exclusively by diffusion until they become revascularized[6]. Therefore, new strategies aiming at protecting the islets from inflammatory insults and/or promoting graft revascularization may be effective for improving clinical islet transplantation outcomes.

Recent studies have demonstrated the functionality of three-dimensionally assembled β-cell aggregates, or multicellular islet spheroids[7,8]. Modulating the cell composition by combining different cell types of islet spheroids leads to improvement of function and viability due to heterotypic cell–cell interactions and reproduction of the complex natural morphology of the islet[9,10]. This strategy could be brought further, by generating multicellular hybrid organoids consisting of several cell types, i.e., endocrine cells for regulated hormone release, and other cell types with cytoprotective and immunomodulatory properties with the aim to increase islet survival and function after transplantation.

Over the last decades, human amniotic epithelial cells (hAECs) have gained interest in regenerative medicine due to their high proliferative capacity, multilineage differentiation, ease of access, and safety[11]. hAECs express surface markers found on human embryonic stem cells and secrete considerable amounts of proangiogenic and anti-inflammatory growth factors, including vascular endothelial growth factor (VEGF), basic fibroblast growth factor (bFGF), angiogenin (ANG), insulin-like growth factors (IGF), and their binding proteins (IGFBPs)[12–14]. In addition, hAECs secrete high levels of hyaluronic acid, which suppresses tumor growth factor β (TGFβ)—a potent profibrogenic cytokine[15]. Decreased levels of TGFβ expression were observed after AEC transplantation in mice with bleomycin- and $CCl_4$-induced lung or liver injury[16,17]. This growth factor secretion profile and antifibrotic properties make hAECs attractive cells for a construct designed to enhance the engraftment and vascularization of islet cells.

In this study we successfully generated viable and functional insulin-secreting organoids composed of hAECs and dissociated islet cells (ICs) and have shown that incorporation of hAECs into islet-cell constructs markedly enhances engraftment, viability and graft function in model of cell therapy for type 1 diabetes.

## Results

### Characterization of hAECs.
After initial seeding, hAECs rapidly formed proliferating clusters and grew within 5 days into a confluent cobblestone-shaped monolayer (Fig. 1a). After culture, hAECs were characterized by flow cytometry and were positive for epithelial (CD326), mesenchymal (CD90, CD105), embryonic stem-cell (SSEA-4) and pluripotency (Oct-4) markers. Most importantly, the hAECs expressed non-classical class Ib histocompatibility antigens HLA-G and HLA-E (Fig. 1b, c, Supplementary Fig. 8). Human amniotic epithelial cells were negative for hematopoietic cell markers CD34, CD31, and CD45 (Fig. 1b). Our results are consistent with previously reported findings[14].

### Generation and in vitro assessment of organoids.
Figure 2a describes the process used herein to generate islet organoids by mixing ICs and hAECs. Both IC- and IC-hAEC aggregates formed round-shaped spheroids of uniform size with well-defined

smooth borders within 5 days (Fig. 2b). The average diameter of IC-hAEC organoids and IC spheroids were 139 ± 4 μm and 202 ± 2 μm (data are mean ± SEM, $n = 12$), respectively (Fig. 2c). Although ICs and hAECs formed small monocellular islands within the organoid, most ICs were in contact with both cell types as shown by confocal laser scanning microscopy (Fig. 2d). No evidence of cell loss was detected. There was no significant difference in cell viability between IC spheroids and IC-hAEC organoids (Supplementary Fig. 1).

On day 5, harvested spheroids were assessed for insulin expression by qPCR analysis, which demonstrated that insulin mRNA expression was significantly upregulated in the IC-hAEC organoids as compared with IC spheroids (Fig. 2e).

The functionality of the spheroids was evaluated by glucose-stimulated insulin secretion (GSIS) assay. IC-hAEC organoids released considerably more insulin in response to high-glucose as compared with IC spheroids and showed significantly higher SI than controls (4.2 ± 0.4 vs 2.8 ± 0.3, data are mean ± SEM, $n = 5$). These data demonstrate that incorporation of hAECs into islet-cell constructs enhances β-cell function.

### Organoids maintain function after hypoxic stress in vitro.
Both IC-hAEC organoids and IC spheroids were cultured under hypoxic conditions (1% oxygen and 5% $CO_2$ at 37 °C) for 16 h to mimic the ischemic condition taking place in vivo in the early phase of engraftment[18]. This allowed us to examine whether hAECs were able to confer cytoprotection and help to maintain the functional capacity of ICs under ischemic stress. Incubation under hypoxia rapidly caused fragmentation of IC spheroids and increased cell death. By contrast, considerably fewer dead cells were observed within IC-hAEC organoids (Fig. 3a). As anticipated, glucose-induced insulin secretion of monocellular spheroids was seriously impaired. By contrast, SI of the IC-hAEC organoids in GSIS assay was significantly higher (Fig. 3b; Supplementary Table 1). These results were further strengthened by qPCR analysis, which showed about threefold higher insulin mRNA expression in IC-hAEC organoids as compared with controls (Fig. 3c).

To assess the possible molecular mechanisms behind the protective effect of hAECs on hypoxia-induced cell death and dysfunction, expression of HIF-1α, a key regulator of cell response to hypoxia was analyzed in spheroids. Immunofluorescence demonstrated a significant increase in the nuclear localization of HIF-1α in IC + hAEC organoids approaching 50% compared with IC spheroids after exposure to hypoxia (Fig. 3d, e). The higher nuclear HIF-1α expression in IC-hAEC was correlated with a downregulation of the apoptotic genes Casp3, Casp8, and Casp9 and twofold upregulation of the antiapoptotic gene Bcl2 (Fig. 3f) compared with IC spheroids alone. Taken together, these results suggest that the hAECs protect islet cells from ischemia-induced apoptotic injury and help to maintain glucose responsiveness through the upregulation of HIF-1α expression.

### Islet organoid transplantation improves diabetes reversal.
To assess whether incorporation of hAECs into the islet organoids could enhance engraftment and lead to better glycemic control, diabetic SCID mice were transplanted with a marginal mass of 150 IC-hAEC organoids (IC-hAEC group, $n = 25$), IC spheroids (IC group, $n = 25$), or hAECs spheroids (hAEC group, $n = 5$). Mice transplanted with IC-hAEC organoids exhibited enhanced glycemic control, compared with mice grafted with IC spheroids (Fig. 4a). The average nonfasting blood glucose concentrations of mice in the IC-hAEC group were considerably lower than those in IC group at 1 month after transplantation (7.9 ± 1.1 mmol/l for

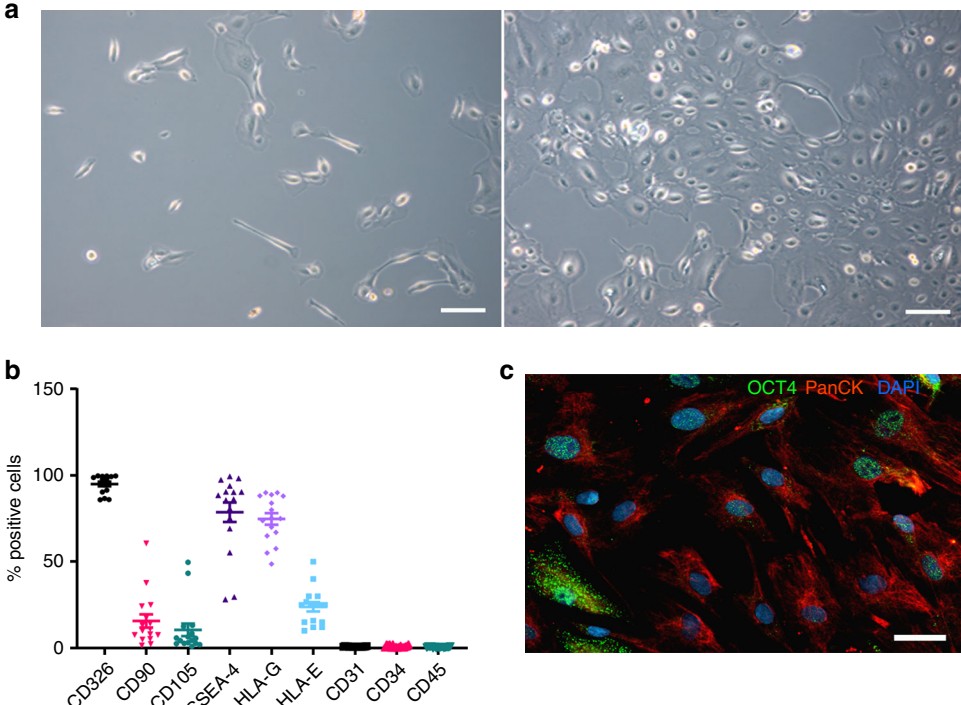

**Fig. 1** Characterization of hAECs. **a** Phase-contrast microscopic images of cultured hAECs at days 2 (left panel) and 5 (right panel) after seeding. Scale bar = 100 μm. **b** hAECs were characterized for various surface markers by FACS. Data are the means ± SEM, of cells obtained from sixteen different donors labeled with specific antibodies (% positive cells). **c** Immunohistochemical detection of cytokeratin (red) and OCT-4 (green) in cultured hAECs; nuclei are labeled by DAPI (blue). Scale bar = 50 μm

IC-hAEC ($n = 13$) vs $18.4 \pm 2.1$ mmol/l for IC ($n = 10$), data are mean ± SEM, $p < 0.0001$, unpaired Student's $t$ test). The cumulative percentage of animals reaching normoglycemia was 96% in the IC-hAEC group vs 16% in the IC group at 1 month after the transplantation (Fig. 4b). In cured animals, the median time to reach euglycemia was $5.1 \pm 0.1$ days in the IC-hAEC ($n = 24$) group and $30 \pm 9.2$ days in the IC group ($n = 8$) (data are mean ± SEM, $p < 0.0001$, unpaired Student's $t$ test). As expected, mice transplanted with hAEC spheroids remained diabetic. Removal of graft-bearing kidneys at different time points after transplantation led to recurrence of hyperglycemia in all mice within 24 h, indicating that the transplanted spheroids were responsible for normalized glucose levels in cured animals.

To investigate the insulin secretory capacity of the graft in vivo, IPGTT was performed at 4 weeks post transplantation. As shown in Fig. 3c, glucose clearance of mice in the IC-hAEC group ($n = 10$) was similar to that of a nondiabetic control at all time points after glucose loading. By contrast, the IC group ($n = 10$) showed abnormal glucose tolerance. To further support the data obtained from the IPGTT, fasting serum insulin and C-peptide levels were measured in the same animals. Both insulin ($242 \pm 32$ pmol/l in the IC-hAEC group vs $130 \pm 29$ pmol/l in the IC group ($n = 6$), data are mean ± SEM, $p = 0.02$, unpaired Student's $t$ test) and C-peptide ($1140 \pm 43$ pmol/l in IC-hAEC group vs $732 \pm 124$ pmol/l in the IC group ($n = 5$), data are mean ± SEM, $p = 0.01$, unpaired Student's $t$ test) concentrations were significantly higher in the IC-hAEC group. These data demonstrate that incorporation of hAECs into the islet organoids enhances functional capacity of islet cells.

**Organoid transplantation enhances graft revascularization**. To evaluate engraftment and revascularization, graft-bearing kidneys were processed for histology. Immunohistochemical (IHC) staining showed larger β-cell mass, as assessed by the insulin-

positive area per field in the IC-hAEC group compared with that of the IC group (Fig. 4d, e) at 120 days posttransplant. This finding was further confirmed by qPCR analysis, which demonstrated that insulin mRNA expression levels were significantly higher (in the IC-hAEC group (Fig. 4f). Similarly, more glucagon and somatostatin-positive cells were found by IHC in the removed grafts of IC-hAEC group compared with grafts of IC group (Fig. 5a–c).

To investigate whether incorporation of hAECs into the islet organoids promotes the process of revascularization, histological sections of the graft-bearing kidneys, harvested at different time points were processed for CD34 and CD31 immunostaining. Higher CD34 and CD31 staining on histological section was observed in IC-hAEC group compared with IC group (Fig. 6a, Supplementary Fig. 2). After quantification, CD34 staining was shown to be 2–4-fold higher in IC-hAEC group compared with IC group (Fig. 6b–d). As expected, in both groups, CD34 staining was higher at day 28 compared with day 14.

**Mechanisms of improved graft function and revascularization**. To elucidate the underlying mechanisms by which hAECs contribute to improved revascularization and function of the graft, we assessed possible differences in VEGF-A production between IC-hAEC and IC groups. Immunohistochemical staining showed higher expression of VEGF-A in the IC-hAEC group compared with in the IC group (Fig. 7a). After quantification. VEGF-A staining in the graft was three times higher in IC-hAEC compared with IC group (Fig. 7b). To assess whether the hAECs stimulated production of proangiogenic factors by the islet cells, rat-specific VEGF-A mRNA levels were measured in IC spheroids and IC-hAEC organoids. IC-hAEC organoids expressed considerably more VEGF-A mRNA than islet IC spheroids (Fig. 7c). Moreover, we examined if endothelial cells in the graft tissue originated from hAECs. To this end, histological sections were exposed to an anti-

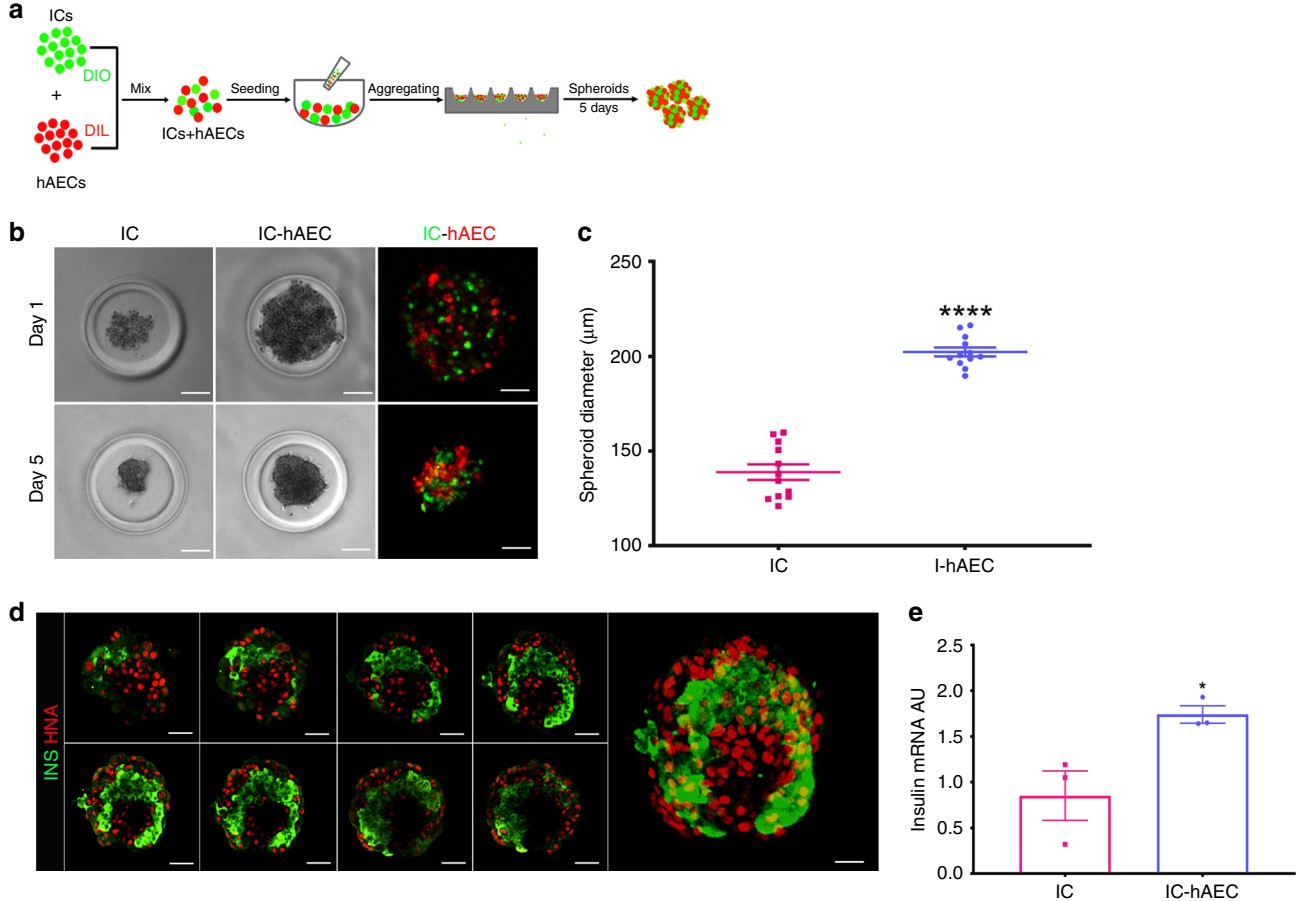

**Fig. 2** In vitro characterization of islet organoids. **a** Schematic representation of islet organoid engineering. After labeling with Dio and DiI, ICs and hAECs were mixed, seeded and incubated several days on 3D agarose-patterned microwells to generate islet organoids. **b** Phase-contrast and corresponding fluorescence views of spheroids in one microwell on days 1 and 5. After 5 day culture (bottom), cells undergo compaction and spheroids appear to acquire a smooth border as compared with aggregated cells at day 1. Scale bar = 50 μm. **c** Diameters of IC spheroids and IC-hAEC organoids ($n = 12$). ****$p > 0.0001$, unpaired Student's $t$ test. **d** Confocal views of islet-cell construct. Cell arrangement and composition of the islet organoid on day 14. Islet-derived cells stained for Insulin (green) and hAECs for human nuclear factor (red). Every ninth section of a $z$-stacked and the entire 3-D reconstructed islet heterospheroid (right panel) are shown. Scale bar = 50 μm. **e** Insulin mRNA expressed by IC spheroids and IC + hAEC organoids; insulin mRNA was analyzed by qPCR, arbitrary units (AU) after normalization to housekeeping genes. *$p < 0.04$, unpaired Student's $t$ test, $n = 3$. All data shown are mean ± SEM

human specific CD31 antibody instead of the anti-rodent CD34 or CD31 antibodies used above. No CD31 staining was found at any time point (Supplementary Fig. 3), suggesting that endothelial cells are not of human origin. These results indicate that human hAECs accelerate the revascularization process mainly by stimulating angiogenic factors in the islet cells, but not through differentiation into the endothelial cells.

We finally examined whether hAEC incorporation into the islet organoids promoted the production of extracellular matrix proteins and adhesion molecules, which are essential to maintain islet morphology and promoting in turn β-cell survival and function[19]. IHC studies demonstrated that expression of collagen IV and laminin (two major basement membrane proteins) was higher in the IC-hAEC group (Fig. 7d–f). Expression of E-cadherin, an adhesion molecule involved in the maintenance of β-cell viability and promoting insulin secretion[20], was considerably upregulated in the IC-hAEC compared with IC group (Fig. 8a, b). These results suggest that incorporation of hAECs into islet-cell constructs enhance basement membrane and production of E-cadherin, thus ensuring proper function of islet cells.

**hAECs remain within grafted organoids over 2 weeks**. To co-localize hAECs within transplanted IC-hAEC organoids, explanted graft-bearing kidneys were stained for anti-human nuclear antigen antibody. Our findings showed that while human-derived cells were abundantly present in the first 2 weeks after transplantation, their number gradually declined over time, and at the end of a 1-month period, only few HNA-positive cells were detectable (Supplementary Fig. 4).

**In vivo function and vascularization of human-derived grafts**. Islet material from two different human donors was used. Each mouse was transplanted with 300 human IC spheroids ($n = 8$) or IC-hAEC organoids ($n = 10$) into the epididymal fat pad. To monitor graft function after transplantation, human C-peptide levels in the blood were measured once a week. As shown in Fig. 9a, C-peptide levels gradually increased after transplantation and were significantly higher in the IC-hAEC group compared with IC spheroid controls. Glucose clearance was studied by IPGTT 1 month after transplantation. IC-hAEC organoids showed normal metabolic function, in contrast with the IC spheroid group (Fig. 9b).

Grafts were removed one month after transplantation and processed for histological analysis to analyze engraftment and vascularization. Histological analysis of explanted grafts from the IC-hAEC group revealed healthy islet morphology. Grafts stained

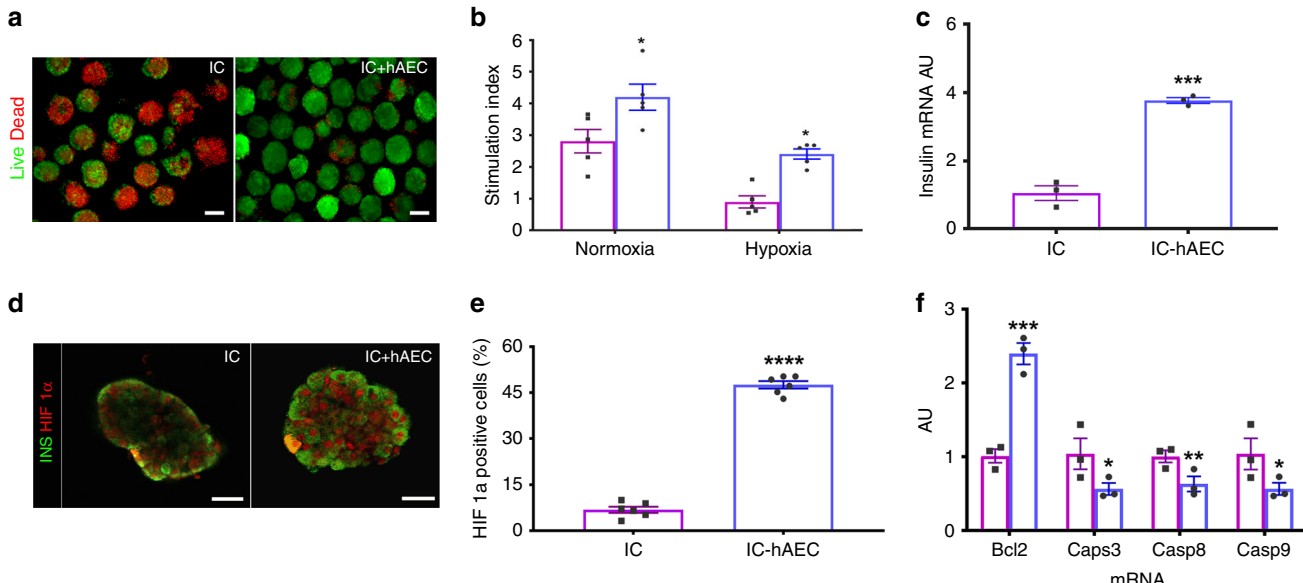

**Fig. 3** Organoid functionality after hypoxic stress. **a** Fluorescence views of IC spheroids and IC-hAEC organoids exposed to hypoxia and assessed for viability by a FDA/PI test; green (FDI) and red (PI) signals indicate live and dead cells respectively. Scale bars = 100 µm. **b** Insulin secretion, expressed as SI, of IC spheroids (magenta columns) and IC-hAEC organoids (blue columns) under normoxic and hypoxic conditions, *$p < 0.03$ and after hypoxic exposure, *$p < 0.02$, two-way ANOVA with Sidak's multiple comparisons test, $n = 5$. **c** Insulin mRNA expressed by IC and IC + hAEC spheroids cultured under hypoxic conditions; insulin mRNA was analyzed by qPCR, and data presented as arbitrary units (AU) after normalization to housekeeping genes, ***$p < 0.0003$, unpaired Student's $t$ test, $n = 3$. **d, e** HIF-1α nuclear localization visualized by immunostaining and its upregulated expression in IC-hAEC spheroids. ****$p < 0.0001$, unpaired Student's $t$ test, $n = 6$. Scale bar = 50 µm. **f** Casp3, Casp8, Casp9, and Bcl2 mRNAs expressed by IC spheroids spheroids (magenta columns) and IC + hAEC organoids (blue columns) cultured under hypoxic conditions; data presented as arbitrary units (AU) after normalization to housekeeping genes. ***$p > 0.0003$, **$p < 0.0075$, *$p < 0.02$, unpaired Student's $t$ test, $n = 3$. All data shown are means ± SEM

positive for insulin, glucagon and for the presence of endothelial cells in newly formed intra-islet micro vessels (Figs. 8c, 9a, b). In contrast, very little graft tissue was retrievable from mice transplanted with IC spheroids. Explants showed extensive loss of islet mass and poor vascularization. Consistent with these results, explanted grafts from the human IC-hAEC group exhibited considerably higher E-cadherin expression levels compared with IC grafts (Supplementary Fig. 5).

These observations demonstrate that integration of hAECs into human islet-cell organoids significantly improves their functionality, viability and vascularization, and confirm the findings obtained with organoids derived from rodent islet cells.

## Discussion
In this study, we have shown that incorporation of hAECs into islet-cell constructs markedly improved secretory function and viability in vitro, in conventional culture and in hypoxic conditions, and engraftment and graft function in vivo. Combined hAEC and islet-cell organoids hold great potential for cell-based therapies for type 1 diabetes. Recent studies have indicated that combining different cell types of cells into hybrid spheroids is a tissue engineering strategy able to provide "building blocks" for larger tissue constructs, with enhanced cell viability, physiologic function and proliferative ability[21]. Thus, the generation of islet-cell-based multicellular spheroids could be an interesting strategy towards the development of novel cell-based therapies in the treatment of diabetes. However, the generation of stable multicellular islet spheroids is quite challenging due to differences in the mode of cellular adhesion of pancreatic islet and other cell types used so far (mostly mesenchymal stem cells)[22].

In this study, we have successfully generated viable and functional islet organoids composed of hAECs and dissociated ICs. We did not observe any segregation of islet cells and hAECs into separate spheroidal units at any stage, as reported by other groups attempting to generate stable hybrid spheroids enriched with mesenchymal stem cells (MSCs)[23,24]. In contrast to what has been reported for MSCs, the even coaggregation of hAECs and islet-derived cells clearly demonstrated by confocal microscopy can be attributed to the epithelial origin of both cell types and the identical mode of cellular adhesion, mediated by the cadherin superfamily[25,26].

Significant islet loss in the early posttransplant period is one of the reasons for suboptimal outcomes of clinical islet transplantation. This occurs, at least in part, by anoikis (programmed cell death secondary to loss of cell-to-extracellular matrix contact) and necrosis caused by ischemia during the revascularization process[27]. Our findings demonstrate a clear protective effect of hAECs on islet cells in hypoxic conditions. While IC spheroids exposed to hypoxia display extensive necrosis and impaired function, IC-hAEC organoids preserve adequate glucose responsiveness and show considerable protection from cell death.

HIF-1α is a transcription factor orchestrating compensatory responses to adapt to hypoxia through modulation of downstream genes involved in angiogenesis and cell survival[28–30]. In islets, HIF-1α upregulates genes involved in glucose metabolism such as glucose transporter 2 (GLUT2) and glucokinase (GCK)[31]. Moreover, recent studies have shown that upregulation of HIF-1α protects islets after transplantation, and thus improves islet transplantation outcomes[32]. In accordance with this, islet bicellular spheroids subjected to hypoxia showed a fivefold increase of HIF-1α expression, which was correlated with a downregulation of the apoptotic genes Casp3, Casp8, and Casp9, and a twofold upregulation of the antiapoptotic gene Bcl2. These findings strongly suggest that hAECs protect islet cells from hypoxic damage through HIF-1α. Moreover, we have observed that hypoxia led to significant loss of E-cadherin expression by IC spheroids (Supplementary Fig. 6), which was correlated with

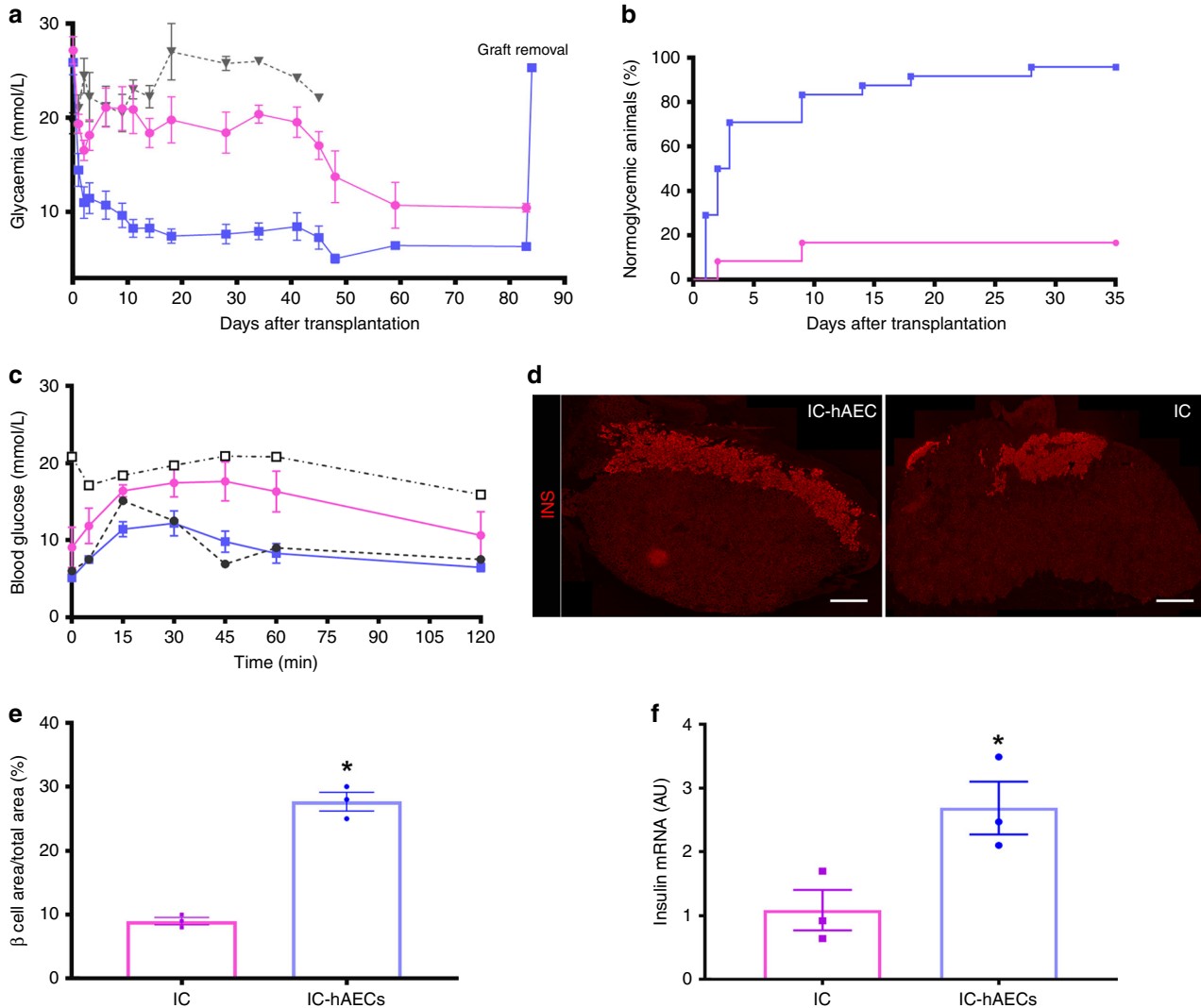

**Fig. 4** In vivo function of islet organoids. **a** Blood glucose measurements. ****$p < 0.0001$ IC-hAEC (blue squares, $n = 25$) vs IC (magenta circles, $n = 25$), *$p$ IC vs, hAEC (gray triangles, $n = 5$), ****$p < 0.0001$ IC-hAEC vs hAEC, one-way ANOVA, with Tukey's multiple comparison test. **b** Percentage of cured mice after islet spheroid transplantation. **c**. Intraperitoneal glucose tolerance tests. **$p < 0.01$ IC-hAEC (blue squares, $n = 10$) vs IC (magenta circles, $n = 10$), one-way ANOVA, with Tukey's multiple comparison test. Gray squares diabetic non-transplanted controls, gray circles non- diabetic, non-transplanted controls. **d**, **e** Insulin-positive area of each group visualized by the immunostaining and its percentage per given area 4 months after transplantation. ****$p < 0.0003$, unpaired Student's $t$ test, $n = 3$. Scale bars = 500 μm. **f** Rat insulin mRNA levels in retrieved grafts after marginal islet spheroid transplantation. Insulin mRNA was analyzed by qPCR, and data presented as arbitrary units (AU) after normalization to housekeeping genes. *$p < 0.01$ vs IC group, unpaired Student's $t$ test, $n = 3$. All data shown are means ± SEM

impaired insulin secretion in response to glucose stimulation. In contrast, IC-hAEC organoids showed preserved E-cadherin expression and adequate in vitro function.

Our in vitro findings were corroborated with in vivo experiments, which demonstrated that transplantation of islet organoids enriched with hAECs resulted in larger β-cell mass engraftment and improved function. Moreover, transplantation of minimal mass of IC-hAEC organoids but not IC spheroids normalized blood glucose levels in STZ-induced diabetic SCID mice. Incorporation of hAECs into the islet-cell constructs accelerated the rate of graft revascularization, which in turn led to a superior engraftment.

Several studies have shown that hAECs are able to promote endothelial cell proliferation and angiogenesis through the secretion of trophic factors[12,13]. Consistent with these data, our findings have demonstrated that hAEC enhance vascular density in the graft at 14 and 28 days after transplantation. Interestingly,

we did not find human-derived endothelial cells in grafts. This finding clearly indicates that hAECs promote revascularization through the stimulation of angiogenic factors in the islet cells, but not through transdifferentiation of epithelial cells. The role of VEGF-A as a key regulator of islet vascularization and function is well known, and a substantial loss of islet vasculature and islet function has been observed in VEGF-A deficient islets[33]. On the other hand, manipulating islets to overproduce VEGF-A accelerates islet revascularization and function after transplantation[34]. Our findings showed a fourfold upregulation of VEGF-A mRNAs in the IC-hAEC organoids compared with IC spheroids suggesting that hAECs mediate the process of neovascularization by stimulating VEGF-A production from the β cells. Stimulation of VEGF-A production could also be attributed to upregulation of HIF-1α triggered by initial hypoxia after transplantation, as VEGF-A is known to be one of the primary target genes regulated by HIF-1α[35]. Enhanced revascularization mediated by hAECs

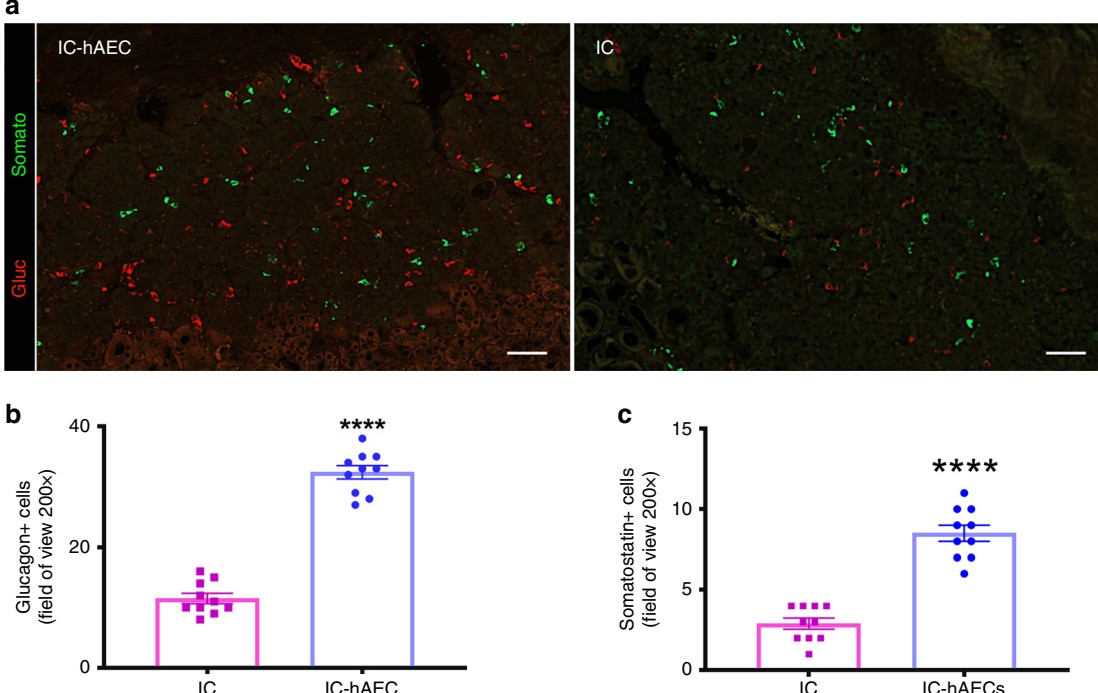

**Fig. 5** Immunohistochemical analysis of hormone production in the grafts. **a–c** Glucagon and somatostatin-positive cells quantified in each group in the field of view (magnification ×200), scale bars 50 µm. ****$p < 0.0001$ vs IC group, unpaired Student's $t$ test, $n = 10$, data are mean ± SEM

contributed to superior engraftment of islet organoids as demonstrated by adequate blood glucose control, glucose tolerance and serum C-peptide concentrations. Preservation of E-cadherin expression, observed in IC-hAEC organoids but not in IC spheroids, could also contribute to the better engraftment and graft function in the presence of hAEC. Direct involvement of E-cadherin in the control of β-cell secretory function in response to glucose was recently demonstrated by Parnaud et al.[36]. Although data on the impact of hypoxic conditions on islet cells during the revascularization period, are lacking, hypoxia has been shown to downregulate E-cadherin expression in other cell types, mostly in the oncology field[37,38]. The mechanisms by which E-cadherin expression was preserved in the presence of hAECs have yet to be studied.

Human amniotic epithelial cells abundantly produce extracellular matrix proteins[39], which have been shown to be essential in promoting β-cell survival and function[19,40,41]. Consistent with reported data, superior engraftment in the IC-hAEC group was accompanied with a substantial increase of Col IV and laminin.

Taken together, these results suggest that the integration of hAECs into islet-cell constructs significantly improves their functionality, viability and engraftment capacity through a variety of mechanisms, including better resistance to ischemia, accelerated revascularization and restoration of cell-to-matrix contacts. Other authors have demonstrated the immunomodulatory properties conferred by amniotic cells[42,43], an observation that we have also made with our hAECs in preliminary experiments (Supplementary Fig. 7). We will next investigate whether these immunomodulatory properties could allow maintenance of the grafts with minimal or no immunosuppression.

This strategy has great potential in the development of cell-based therapies for type 1 diabetes, since the engineering of spheroids into functional mini-organs would allow the exploration of more favorable implantation sites and could be expanded to unlimited sources of insulin-producing cells, such as stem cells or xenogeneic sources.

## Methods

**Antibodies and reagents**. References and catalog numbers of all antibodies and reagents used are listed in Supplementary Tables 2–5.

**Animals**. Male, 6–8-week-old SCID mice and male 8-week-old Sprague-Dawley (SD) rats (250–300 g) were purchased from Janvier Labs (Le Genest St-Isle, France) and kept in the animal facilities at the University of Geneva with free access to food and water. All animal procedures were approved by the University of Geneva Institutional Animal Care and Use Committee.

**Human samples**. Studies involving human tissues were approved by the Commission Cantonale d'Ethique de la Recherche (CCER), in compliance with the Swiss Human Research Act (810.30). Amniotic membranes were obtained from term healthy placentas of women undergoing elective cesarean section, under pre-Basec CCER protocol PB_2017-00101 (14-273), listed at https://www.ge.ch/document/liste-protocoles-soumis-2015. Informed, written consent was obtained from each placenta donor prior to amniotic tissue collection. Human islets isolated from brain-dead multiorgan donors were obtained from the Lille University Hospital. The use of human islets for research was approved by CCER protocol 2016-01979.

**Rat pancreatic islet isolation and dissociation**. Rat pancreatic islets were isolated by enzymatic digestion (collagenase V, Sigma-Aldrich) and purified by centrifugation on a Ficoll density gradient[44]. Purified islets were incubated (37 °C, 5% $CO_2$) 24 h in DMEM medium (ThermoFisher Scientific) supplemented with 10% (v/v) fetal bovine serum (FBS; Merk Millipore), 1 mmol/l sodium pyruvate (Sigma-Aldrich), 11 mmol/l glucose (Bichsel, Interlaken, Switzerland), 0.05 mmol/l 2-mercaptoethanol (Bio-Rad), 2 mmol/l L-Glutamin, 100 U/ml Penicillin and 0.1 mg/ml Streptomycin (1% (v/v) of a L-Glutamin-Penicillin-Streptomycin stock solution from Sigma-Aldrich). Islets were then incubated in 0.05% (w/v) trypsin-EDTA[45] and dispersed into single cells.

**Human pancreatic islet dissociation**. Islets from two separate human islet preparations were used. Prior to dissociation, human islets were cultured in CMRL 1066-medium containing 5.6 mmol/L glucose and supplemented with antibiotics, HEPES, and 10% FBS. Islets were dissociated into single cells[20]. Briefly, islets were rinsed twice with PBS, resuspended in 1 mL Accutase (Innovative Cell Technologies) and incubated at 37 °C with gentle pipetting every 30 s. When dissociation was considered to be complete, cells were resuspended in cold complete CMRL and incubated at 37 °C in nonadherent Petri dishes.

**Isolation of human amniotic epithelial cells (hAECs)**. hAECs were isolated according to the method of Miki et al.[46]. Briefly, the amnion was mechanically peeled from the underlying chorion and washed four times with cold HBSS supplemented

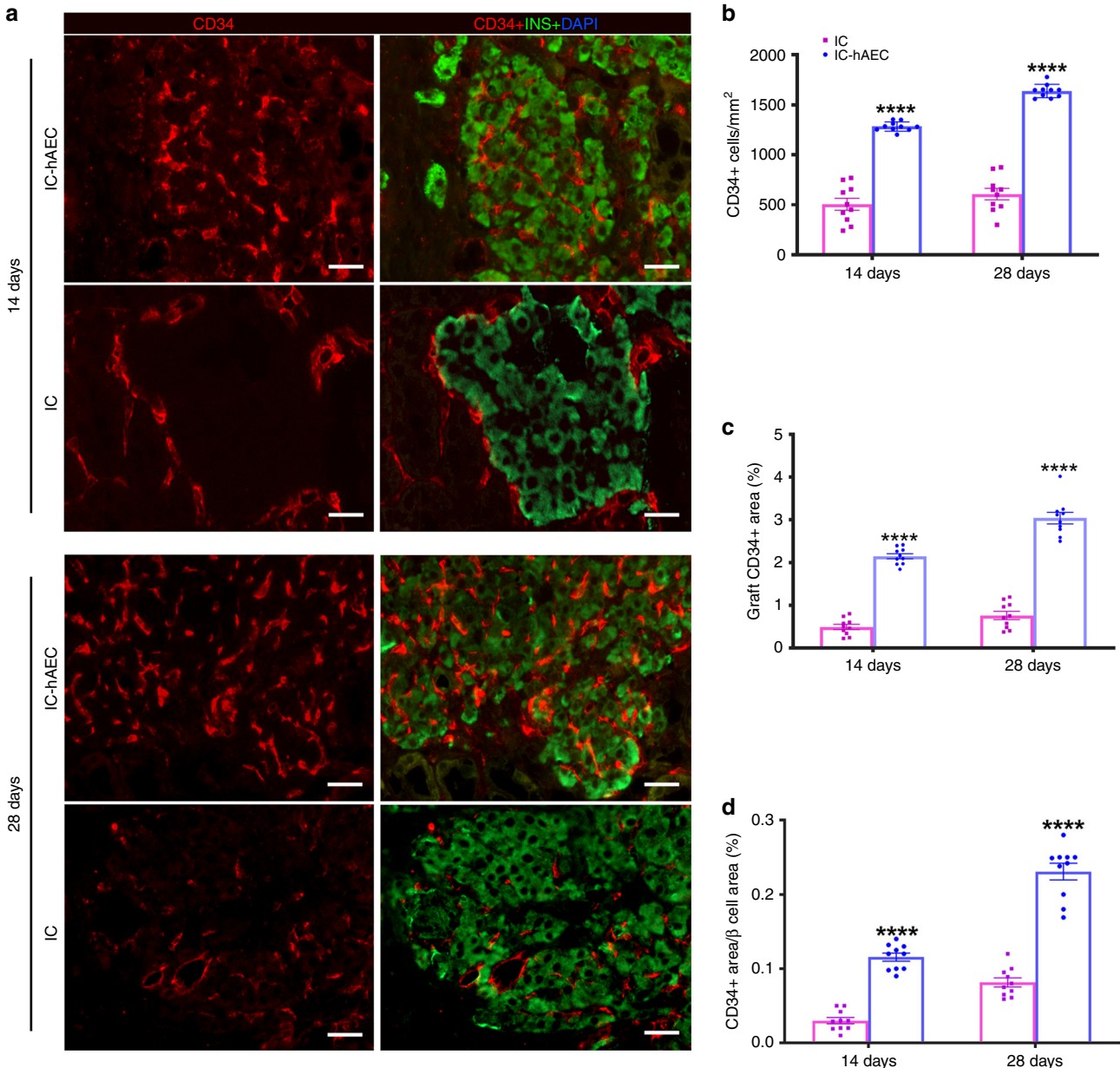

**Fig. 6** Enhanced revascularization of the grafts in IC-hAEC group. **a** The blood vessels of the graft site were detected at day 14 and 28 using CD34 immunostaining. Scale bars = 20 μm. **b–d** The total number of vessels was calculated as the number of endothelial cells per mm$^2$ of the graft. The vessel/graft and vessel/β-cell ratios were calculated as a percent of the graft and insulin-positive area respectively. ****$p < 0.0001$ vs IC group, two-way ANOVA, with Tukey's multiple comparison test, $n = 10$. All data shown are mean ± SEM

with 100 U/ml penicillin, 100 mg/ml streptomycin, and 0.25 mg/ml amphotericin B. To isolate hAECs, the amnion was incubated with 0.05% trypsin/EDTA for 40 min at 37 °C with gentle shaking. Dispersed cells were collected by centrifugation at 500 g and after four washes were plated at a density $20 \times 10^4$ cells/cm$^2$ in DMEM/F-12 medium (ThermoFisher Scientific) supplemented 2 mmol/l L-Glutamin, 100 U/ml Penicillin, and 0.1 mg/ml Streptomycin (1% (v/v) of a L-Glutamin-Penicillin-Streptomycin stock solution from Sigma-Aldrich), 1 mmol/l sodium pyruvate (Sigma-Aldrich), 1% (v/v) MEM NEAA 100X (ThermoFisher Scientific), 0.1% fungin (InvivoGen, San Diego, CA), 10% FBS, 0.05 mmol/l 2-mercaptoethanol (Thermo-Fisher Scientific), 10 ng/ml human recombinant epidermal growth factor (EGF; Sigma-Aldrich) and cultured at 37 °C, 5% CO$_2$ in a humidified atmosphere for 48–72 h to form a confluent monolayer. The culture medium was changed three times a week. hAECs were harvested at 80% confluence by mild trypsinization and were either used fresh or cryopreserved in 90% FBS and 10% DMSO for later use. To generate spheroids hAECs were thawed and cultured for 5 days.

**Characterization of hAECs.** Cultured hAECs were harvested as described above, then rinsed three times with phosphate-buffered saline (PBS)−0.1% BSA

supplemented with 0.01% sodium azide (hereafter referred as PBS–BSA-N$_3$), and aliquots of $10^6$ hAECs in 100 μl of PBS–BSA-N$_3$ were incubated for 30 min at 4 °C with the following fluorescent-conjugated antibodies or matched-isotype control IgGs: FITC-conjugated anti-CD 105, BV421-conjugated anti-CD326, PerCP-Cy5.5-conjugated anti-SSEA-4 (1:50 dilution; all from BD Biosciences), PE-conjugated anti-CD 90 (1:100 dilution; BD Biosciences) and anti-APC-conjugated HLA-G (1:50 dilution, Biolegend), HLA-E (1:50 dilution; Biolegend, London, UK), CD31 (1:30 dilution; Biolegend), CD34 (1:25 dilution; Abcam), and CD45 (1:25 dilution; Biolegend) antibodies. DRAQ7 (1:10 dilution; Biostatus) far-red fluorescing viability staining was used to exclude dead cells. Cells were analyzed by flow cytometry on a Gallios cytometer (Beckman Coulter, Indianapolis, Indiana, USA) using Kaluza Analysis software from Beckman Coulter (Version 1.5.20365.16139).

hAECs cultured on collagen-coated coverslips were fixed in 4% paraformaldehyde (PFA) at 4 °C for 10 min, rinsed twice with PBS and permeabilized with PBS containing 0.5% Triton X-100 for 15 min. After two washes in PBS, slides were incubated in 1% BSA in PBS for 1 h. Samples were incubated overnight at 4 °C with anti-pancytokeratin (1:75 dilution), and anti-Oct-4 (1:200 dilution) antibodies (all from Abcam) in PBS containing 0.2% Triton X-100. Cells were rinsed twice with PBS and incubated 2 h at room temperature with

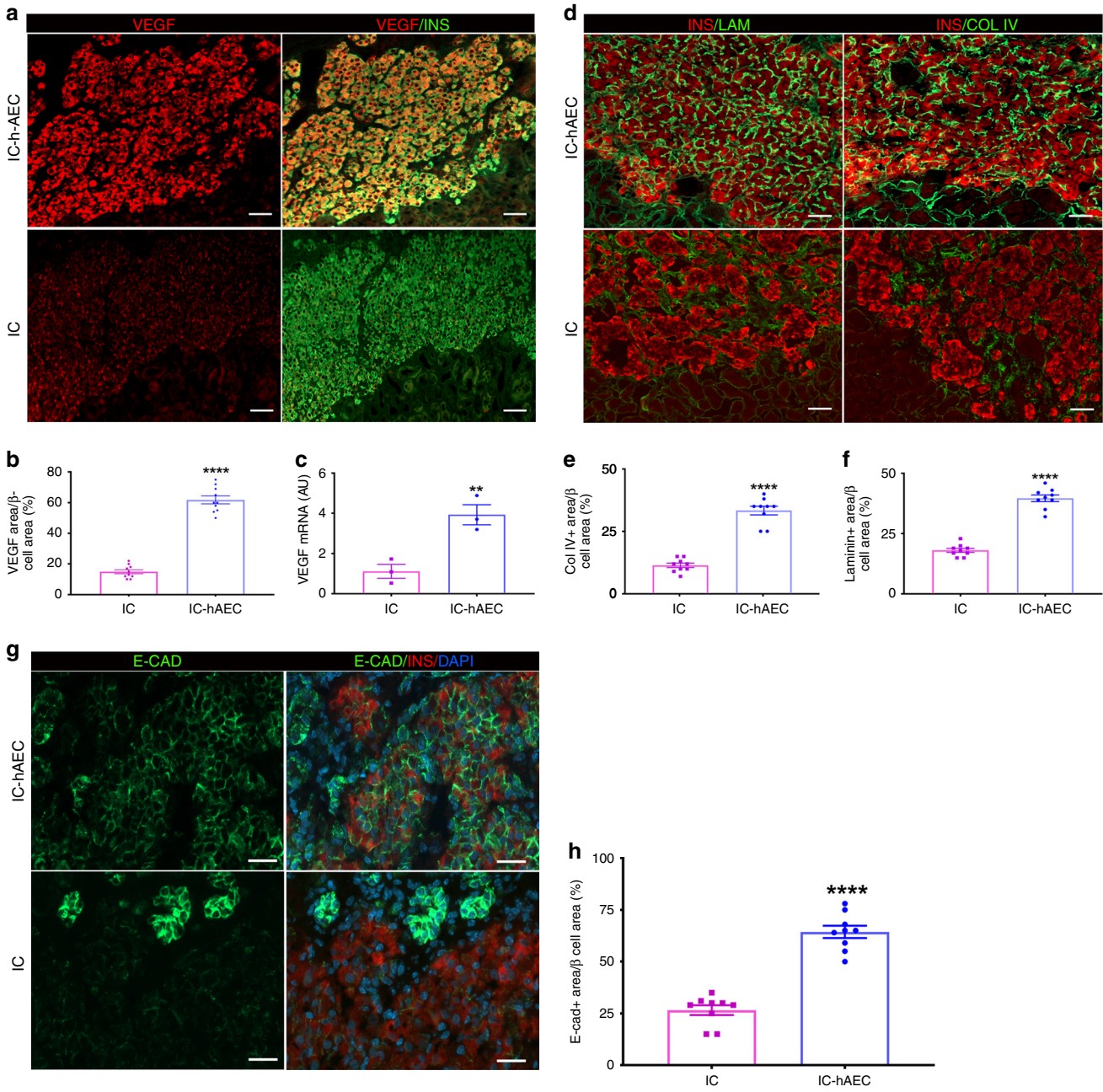

**Fig. 7** The mechanisms of the enhanced vascularization and improved function of the IC-hAEC grafts. **a** The graft-bearing kidneys stained for VEGF-A and insulin at day 14 after transplantation. Scale bars = 100 μm. **b** The mean VEGF-A expression per insulin-positive area, ****$p < 0.0001$ vs IC group, unpaired Student's $t$ test, $n = 10$. **c** VEGF-A mRNA expressed by IC and IC + hAEC spheroids as analyzed by qPCR, data presented as arbitrary units (AU) after normalization to housekeeping genes, with the IC group set to, **$p < 0.001$ vs IC group, unpaired Student's $t$ test, $n = 3$. **d** Representative images of the grafts labeled for collagen IV, laminin and insulin. Scale bars = 100 μm. **e**, **f** The Col IV-positive and the laminin-positive areas were calculated as a percentage of the graft area, ****$p < 0.0001$ vs IC group, unpaired Student's $t$ test, $n = 10$. **g**, **h** Expression of E-cadherin as analyzed by immunohistochemistry in the graft site was considerably upregulated in the IC-hAEC group. Scale bars = 100 μm. ****$p < 0.0001$ vs IC group, unpaired Student's $t$ test, $n = 10$. All data shown are mean ± SEM

FITC- or Cy3-conjugated secondary antibodies. Finally, cells were rinsed with PBS and mounted with aqueous mounting solution containing DAPI (4′,6-diamidino-2-phenylindole; ProTaqs MountFluor Anti-Fading, Quartett Biochemicals, Berlin, Germany) for nuclear counterstaining.

**Generation of spheroids**. Spheroids were generated on 3D agarose-patterned microwells by distributing 500 μl of warm, sterile 2.5% agarose (Promega, Dübendorf, Switzerland) solution into 256-well micromolds, with a well diameter 400 μm (3D Petri Dish; Microtissues Inc., Sigma-Aldrich). After solidification, agarose casts were removed from the molds, transferred into 12-well culture plates, washed three times with PBS and stored at 4 °C until used. Before cell seeding, the agarose casts were equilibrated in culture medium for at least 1 h at 37 °C. To form

monocellular spheroids (IC- and hAEC spheroids) dispersed islet cells (ICs) and hAECs (128,000 cells/cast and 500 cells/spheroid) were seeded alone. Islet organoids were formed by mixing ICs and hAECs at a ratio of 1:1 (128,000 cells/cast and 500 cells/organoid). The number of islet cells was chosen in order to obtain IC spheroids of ~150 μm diameter, i.e., the size of an "islet equivalent" (IEQ).

After cell seeding, agarose casts were centrifuged at 500 g for 5 min to trap the cells into the microwells and cultured for 5 days. Culture medium was changed every other day. Cell aggregation and spheroid formation were observed daily under the microscope. Spheroids were removed from the casts by upside down centrifugation at 300 g for 2 min. The average diameters of spheroids were measured by analyzing light microscopy images using ImageJ software (NIH, Bethesda, MD).

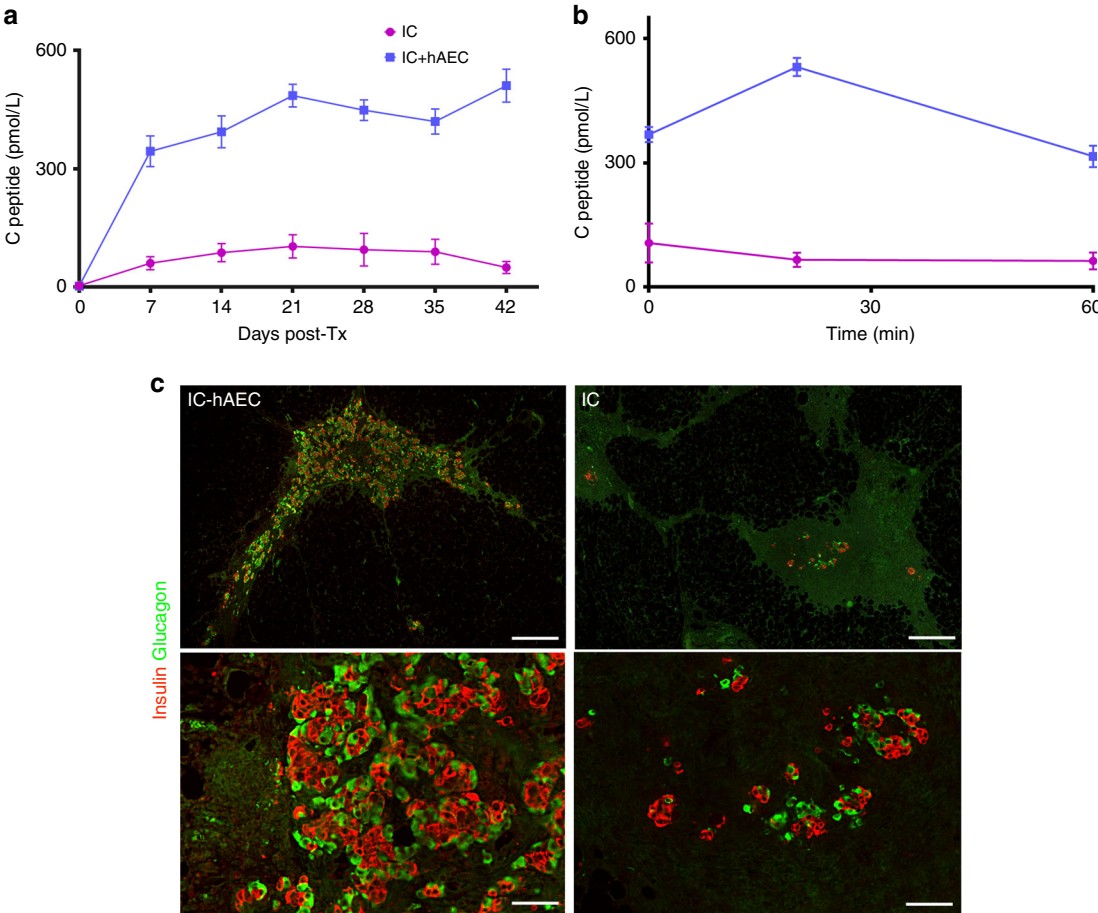

**Fig. 8** Function of a human-derived organoids transplanted in the epididymal fat pad. **a** Human C-peptide measurements of mice transplanted with 300 islet organoids (IC-hAEC group, blue squares, $n = 10$), or with 300 islet-cell spheroids (IC group; magneta circles, $n = 8$) ***$p < 0.0007$, unpaired Student's $t$ test. **b** Human C-peptide levels after intraperitoneal glucose challenge 4 weeks after transplantation. Magneta circles: IC, blue squares: IC-hAEC. **$p < 0.008$, unpaired Student's $t$ test, $n = 5$. **c** Representative images of the graft stained for insulin (red) and glucagon (green). Scale bar, upper panel 500 μm, lower panel 50 μm

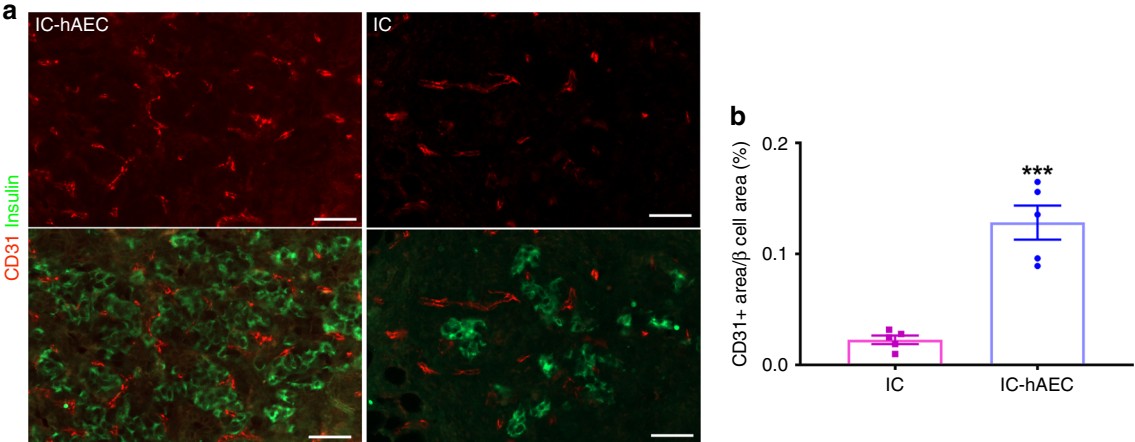

**Fig. 9** Revascularization of the grafts transplanted in the epididymal fat pad. **a** Fluorescent images of the grafts stained for blood vessels (CD31, red) and insulin (green). Scale bar = 20 μm. **b** Percentage of vessels per insulin-positive area. ***$p < 0.0002$ vs IC group, unpaired Student's $t$ test, $n = 5$. All data shown are mean ± SEM

In order to visualize cell distribution in islet organoids, ICs and hAECs were prelabeled with fluorescent carbocyanine dyes CM-DiI (red) or CM-DiO (green) (ThermoFisher Scientific), according to the manufacturer's protocol. Labeled cells were visualized using a spectral confocal microscope (Nikon A1R; Nikon Imaging, Egg, Switzerland).

**Cell viability and functional assessment of IC-hAEC organoids**. To examine whether hAECs are able to express cytoprotective abilities and help to maintain functional capacity of islet cells under ischemic stress, rat IC spheroids and IC-hAEC organoids were cultured under hypoxic conditions (1% oxygen and 5% $CO_2$ at 37 °C) for 16 h.

Cell viability was assessed by fluorescein diacetate (FDA) and propidium iodide (PI) staining. To assess viability, spheroids were incubated in fluorescent dyes for 5 min in the dark and then imaged under an epifluorescent microscope (Leica Microsystems, Heerbrugg, Switzerland). Viable cells were stained in green (FDA), while DNA of dead cells was stained in red (PI). Cells were photographed and using the ImageJ software at least 100 organoids per condition were analyzed to determine the percentage of FDA positive cells (viable cells).

To assess insulin secretory capacity in response to glucose stimulation, 50 IC- and IC-hAEC organoids were handpicked in triplicate, transferred to an ultra-low attachment plate and preincubated for 1 h in Krebs–Ringer buffered HEPES (pH 7.4) with 0.1% (wt/vol) BSA containing 2.8 mmol/l glucose. Organoids were then incubated at 37 °C for 1 h in low-glucose (2.8 mmol/l) and high-glucose (16.7 mmol/l) solutions, subsequently. Supernatants were collected and stored at −20 °C. Insulin concentration in supernatants (secreted) was measured using a rat insulin ELISA kit (Mercodia, Uppsala, Sweden) and normalized to the total insulin content of the corresponding organoid lysates. The ability of organoids to secrete insulin in response to glucose was define as the ratio of the insulin secreted in high-glucose to the insulin secreted in low-glucose condition, hereafter referred as the stimulation index (SI).

**Immunohistological analyses of cultured spheroids**. At day 5 after culture, recovered spheroids were fixed in 4% (w/v) PFA for 30 min at room temperature. Following fixation, spheroids were washed three times in PBS, permeabilized in 0.5% Triton X-100/PBS for 4 h and incubated in 0.5% BSA/0.1% Triton X-100/PBS at 4 °C for 1 h to block unspecific sites. Blocking was followed by incubation with primary antibodies against insulin (1:100 dilution, DakoCytomation, Baar, Switzerland), glucagon (1:4000, Sigma-Aldrich), E-cadherin (1:50; Cell Signalling Technology, Danvers, MA), human nuclear antigen (1:200; Lifespan Biosciences, Seattle, WA), and HIF1A (1:100; Abcam) used in combination as indicated in the Results section. The secondary antibodies used were goat anti-mouse, anti-rabbit, anti-guinea pig (1:300; ThermoFisher Scientific), or goat anti-guinea pig (1:200; Jackson ImmunoResearch Laboratories, Rheinfelden, Switzerland). Both primary and secondary antibodies were diluted in 1% BSA/0.1% Triton X-100/PBS and the incubations were carried out overnight at 4 °C. Spheroids were then transferred to Ibidi microscopy culture chambers (Ibidi, Planegg, Germany) and subjected to optical sectioning 1-µm increments in axial (z) dimension using a spectral confocal microscope (Nikon Imaging).

**Diabetes induction and xenogeneic islet transplantation**. One week before transplantation mice were rendered diabetic by single intraperitoneal dose of 250 mg/kg streptozotocin (Sigma-Aldrich). Nonfasting blood glucose was measured daily after streptozotocin injection from blood samples taken from the tail of the animal using a portable glucometer (Freestyle Precision, Abbott Diabetes Care, Baar, Switzerland). Only mice with blood glucose levels > 20 mmol/l for at least 3 consecutive days were used as recipients.

For experiments using rat islet cells, 150 IC spheroids and IC-hAEC organoids were handpicked, packed in PE50 tubing (PhyMep, Paris, France) and transplanted under the kidney capsule of diabetic mice using a screw-drive syringe (Hamilton, Reno, NV). Nonfasting blood glucose was measured daily during the first week and twice weekly thereafter. Graft function was defined as blood glucose levels below 11.1 mmol/l. To exclude residual or recovery function of the native pancreas and to ascertain rapid loss of euglycemia, graft-bearing kidneys were removed 5, 14, 30, and 120 days after transplant and blood glucose levels were monitored.

In another set of experiments, 300 human IC spheroids and IC-hAEC organoids were transplanted into the epididymal fat pad (EFP) of SCID mice for assessment of survival, function and vascularization of human islet-cell-derived organoids. The human omentum was recently shown to be a relevant extrahepatic site for clinical islet transplantation[47], and the epididymal fat pad is considered the murine equivalent of the human omentum for islet transplantation[48].

**In vivo functional assessment of transplanted islet organoids**. To assess glucose responsiveness, intraperitoneal glucose tolerance test (IPGTT) was performed one month after islet transplantation. Briefly, after a 6-h fast, mice were injected intraperitoneally with 2.0 g/kg glucose and blood glucose measured at 0, 15, 30, 45, 60, and 120 min.

**Immunohistological analyses of transplanted islet organoids**. Retrieved grafts were fixed in 4% (g/vol) PFA and embedded in paraffin or cryopreserved at −80 °C in optimum cutting temperature (OCT) compound (Tissue-Tek, Sakura Finetek, Tokyo, Japan). Tissue samples were serially sectioned and 10 randomly picked 5 µm sections per graft were used for further analysis. Sections were processed for hematoxylin-eosin staining or immunofluorescence. Tissue samples were permeabilized in 0.5% Triton X-100/PBS for 10 min, incubated in 0.1% BSA in PBS, pH 7.4 (PBS) for 30 min at room temperature to block nonspecific sites and incubated with primary antibodies against insulin (1:100; DakoCytomation), glucagon (1:4000; Sigma-Aldrich), somatostatin (1:100; DakoCytomation), human specific CD31 (1:50; DakoCytomation), CD31 (1:50; Abcam, crossreacting with human, pig and mouse), CD34 (1:2500; Abcam), E-cadherin (1:50; Cell Signalling

Technology), HIF1α (1:100; Abcam), human nuclear antigen (1:200; Lifespan Biosciences), VEGF (1:200; Abcam), Laminin (1:30; Sigma-Aldrich), and Collagen IV (1:30; Bio-Rad, Basel, Switzerland) in PBS with 0.1% BSA serum for 16–18 h at 4 °C. Afterward, samples were washed twice in PBS and incubated in goat anti-mouse, anti-rabbit, anti-guinea pig (1:300; ThermoFisher Scientific), or goat anti-guinea pig (1:200; Jackson ImmunoResearch Laboratories) secondary antibodies for 1 h. Histological slides were mounted with aqueous mounting medium containing DAPI (ProTaqs MountFluor Anti-Fading, Quartett Biochemicals, Berlin, Germany) for nuclear counterstaining

The entire images of 20 serial sections were captured using a Zeiss Axioscan. Z1 slide scanner (Zeiss, Feldbach, Germany) for automated imaging. Morphometry and fluorescent analysis were performed using ImageJ software.

**Real-time quantitative PCR analysis**. Transplanted spheroids were removed from the retrieved grafted kidneys using microscissors[7].

Briefly, RNA was extracted from spheroids or microdissected grafts using the RNeasy minikit (Qiagen, Courtaboeuf, France) and reverse transcribed using the High Capacity cDNA Reverse transcription kit (ThermoFisher Scientific). Gene amplification was achieved by RT-PCR using either the TaqMan Fast Advance Master Mix (ThermoFisher Scientific), or the Takyon No-Rox SYBR Core Kit blue dTTP (Liège, Belgium). Primers used for amplification were rat INS2 (Rn01774648-g1), rat BCL2 (Rn99999125-m1), rat CASP3 (Rn00563902-m1), rat CASP8 (Rn00574006-m1), rat CASP9 (Rn00581212-m1), rat VEGF (5′-AAC GCG AGT CTG TGT TTT TGC-3′), rat ACTB (Rn00667869-m1), and rat RPLP1 (5′-TCT CTG AGC TTG CCT GCA TCT ACT-3′). Gene expressions values were normalized to the housekeeping genes (Actb and Rplp1), and calculated based on the comparative cycle threshold Ct method ($2^{-\Delta Ct}$ method).

**Statistical analysis**. All experiments in this study were reproduced with similar results for at least three times. Continuous and categorical variables are presented as mean ± SEM. Comparisons between two groups were performed with unpaired Student's $t$-test or one-way ANOVA when appropriate. All statistical analyses were performed with Prism software 7.02 (GraphPad, La Jolla, CA, USA), and $p < 0.05$ was considered statistically significant.

**Reporting summary**. Further information on research design is available in the Nature Research Reporting Summary linked to this article.

## Data availability

The source data underlying the main Figs. 1b, 2c, e, 3b, c, e, f, 4a, c, e, f, 5b-c, 6b-d, 7b-c, e-f, h, 8a-b and 9b are provided as a Source Data file.

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

## Acknowledgements

This work was supported by grants from the Swiss National Science Foundation (Grant #310030_173138, to T.B., E.B., and D.B.), from the Fondation Privée des HUG ("CONFIRM Project" to T.B.), from the Fondation Romande de Recherche sur le Diabète (to E.B. and T.B.) and from the European Foundation for the Study of Diabetes (to E.B.). Human islets were provided thanks to a grant from the Juvenile Diabetes Research Foundation (JDRF grant #31-2012-783).

## Author contributions

E.B. conceived the idea, conceptualized and supervised the study. E.B. and T.B. initiated and designed the experiments. F.L., V.L., R.B., V.K., L.P., K.B., D.G. and C.H.W. performed the experiments. J.K.C. isolated and provided human islets. M.B. provided human placentas. V.L., F.L., A.F., D.B., T.B. and E.B. analyzed the data. All authors provided input for the paper writing and discussion.

## Competing interests

The authors declare no competing interests.
