## [Peer Review File · Nature Communications]

Reviewers' comments:

Reviewer #1 (Remarks to the Author):

This is an excellent study, reporting a cell-based strategy to improve islet cell engraftment and function for the treatment of diabetes mellitus. The strategy is based on the combination of islet cells with amniotic epithelial cells. In vitro studies showed that this combination led to a marked improvement in the viability in hypoxic conditions and in the insulin-secretory function of islet beta cells. In vivo studies were performed with rat islet cells and human amniotic epithelial cells (hAECs), transplanted in immunodeficient SCID mice rendered diabetic with streptozotocin. In this animal model, the combination of islet cells with amniotic epithelial cells determined a dramatic and long-term improvement of glycemia. Neither islet cells alone nor amniotic epithelial cells alone yielded such benefit. The glucose lowering effects were observed with the transplantation of extremely small amounts of islet cells, pre-aggregated and co-transplanted at a 1:1 ratio with amniotic epithelial cells. In vivo effects were attributed to an increase in beta cell engraftment, an increase in graft vascularization, and a restoration of cell-to matrix contact in the graft.

These claims are novel and could have a significant impact in the field of cell-based therapies for diabetes mellitus.

The in vivo results are striking: the authors report diabetes reversal in the vast majority of the recipients within 10 days after transplantation with an extremely low number of islet cells (150 organoids of islet cells and hAECs -- each composed by approximately 250 islet cells and 250 amniotic epithelial cells, for a total of approximately 37,500 islet cells and 37,500 amniotic epithelial cells per animal).

An amount of islet cells that was double than that (150 islet cell organoids, each composed by 500 islet cells, for a total of approximately 75,000 islet cells) did not determine glycemia normalization in the vast majority of the recipients for almost two months.

There is one point that deserves to be further investigated in future studies: what is the fate of the transplanted human AECs, when co-transplanted with islets? The persistence of human AECs in the grafts could be further investigated, as well as their eventual maturation into insulin-producing beta-like cells. The investigators utilized an anti-human nuclear antigen antibody for the initial experiments of in vitro aggregation (see Figure 2D), but did not report results with such antibody on the explanted grafts. Moreover, the in vivo release of human insulin/C-peptide, which can be differentiated from rat and mouse insulin/C-peptide via human-specific antibodies, could be monitored.

Minor comments:

Results/Characterization of hAECs:

HLA-G and HLA-E belong to the non-classical HLA-class Ib family.

Results/Generation of spheroids and assessment of in vitro viability and functionality: Here and in Figure 2F the authors report the Stimulation Index (SI) [i.e. the ratio of insulin amount quantified after exposure to high glucose concentration, divided by the amount quantified after exposure to low glucose concentration]. Also the absolute amounts of insulin would have been informative, especially if normalized by DNA content, and could have strengthened the results.

Results/In vivo functional assessment of transplanted islet organoids:

Please indicate the n of mice that underwent IPGTT, for each group of animals.

Results/Transplantation of islet organoids preserves endocrine cell mass and enhances graft revascularization:

Here and in Figure 5 the authors indicate that CD34 immunostaining was used to analyze vascular endothelial cells, whereas in the Methods section (Immunohistological analyses of transplanted islet organoids) they indicate a CD31 antibody and not a CD34 antibody. Please clarify.

Results/Underlying mechanisms for improved graft function and revascularization: "but not through differentiation into the epithelial cells" ...do the author refer to endothelial cells?

Discussion

State "Mesenchymal Stem (or stromal) Cells" before the acronym "MSC".

Materials and Methods:

To enable replication of the experiments, please indicate the Catalog Numbers of the products used.

Materials and Methods/Isolation of human amniotic epithelial cells (hAECs)

Please indicate the freezing medium utilized for cryopreservation, and indicate whether or not cryopreserved cells were used in in vivo experiments.

Materials and Methods/Characterization of hAECs:

The details of Oct-4 and HLA-E antibodies are missing.

Materials and Methods/Cell viability and functional assessment of IC-hAEC organoids:

How many clusters were handpicked?

Materials and Methods/Diabetes induction and xenogenic islet transplantation:

Please indicate here the n of spheroids handpicked.

Figure 5 and its Figure Legend:

Please define more clearly the units of measure in the Y axes (This reviewer understands: cell number/mm² in B; % of graft area in C, and % of beta cell area in D). Please maintain consistency between the annotations in the Figures and those in the Figure Legend.

Figure 6D

The results presented in Figure 6D and 6E don't fit together. Are the columns of Figure 6D mis-labeled? Also, Please make the whole Figure 6 consistent, e.g. markers in columns - and cell types in rows.

Reviewer #2 (Remarks to the Author):

The manuscript by Lebreton and colleagues describes an innovative approach to pancreatic islet transplantation whereby engrafted beta cell survival, function, and vascularization are augmented via the incorporation of human amniotic epithelial cells into re-aggregated pseudo-islet spheroids. The work is of sound technical quality. The use of hAEC as a cell type to improve grafted islet function is novel; however, it is not clear how the end results are demonstrably different or improved over other similar co-culture approaches using, for example, MSCs (e.g. DOI's: 10.2337/db16-1068, 10.1371/journal.pone.0073526, 10.1007/s00125-011-2053-4, 10.1007/s00125-016-4120-3). The results of this study are indeed convincing for the most part, and would be interesting to the community of scientists who study islet transplant re-vascularization. However, the scope of the implications of this study for human islet transplantation are somewhat limited by the choice of the in vivo model. The study focuses mostly on transplantation outcomes, so there is only a peripheral emphasis on mechanisms (HIF-1, etc). The impact of the paper could be improved by providing a stronger translational approach that goes beyond the present results generated with rat islets transplanted into STZ-induced SCID mice.

I have the following concerns / questions:

Major:

1. Please provide evidence of the fate of amniotic epithelial cells post-transplantation. How long do they survive in the islet graft? (Figure 6A is suggestive that there do not appear to be any cells other than beta cells). Or, do the AECs continue to proliferate in vivo and is there a danger of tumorigenesis in immune compromised or immune suppressed individuals?
2. To increase the broader impact of the work, it would be critical to demonstrate the efficacy of hAEC-islet organoids using primary human islets or renewable sources of human beta cells. This is because there are significant species differences between rodent and human islets in terms of vasculature, tolerance to hypoxia, and long-term graft survival.
3. The kidney capsule is a high-performing site for rodent islet transplantation but has limited potential use for human clinical islet transplantation. This is due to the limited subcapsular space that may be insufficient for placing enough human islets to control glycemia, the potential for pre-existing kidney disease in severely diabetic patients, and the risk of inducing damage to kidneys from the graft. The omentum and subcutaneous spaces have more impact for human translatability. The broader importance of the hAEC approach could be better understood in a more translatable graft location.
4. The authors suggest that hAEC may have immunosuppressive properties leading to islet graft immune acceptance without systemic immunosuppression. Data testing this hypothesis would significantly elevate the impact of this study. Lacking such data, the demonstrated improvements to vascularization and insulin secretion alone are less broadly impactful as these metrics have been addressed before through many different strategies.

Minor:

5. Figure 3A-F appears to be lacking an important control: the non-hypoxic condition for IC and IC+hAEC. Please include these key data for comparison.
6. Figure 4: From the resolution of the provided images, it is not totally clear that the CD34+ staining marks actual blood vessels and is not, rather, the hAEC cells. This ambiguity could be addressed by showing staining with a second marker such as CD31, vWF, or tomato lectin perfusion (can show vessel patency). Supplementary data should also be included to validate both antigen and species specificity of the anti-human and anti-rodent CD34 antibodies if conclusions are based solely on this marker.
7. Data representation: please show individual data points throughout rather than bar charts, especially for experiments with small sample sizes.
8. I counted two instances of the phrase "data not shown." Rather than excluding these data, please include it as supplementary information.
9. Figure 6G: The total lack of E-cadherin staining in islet-alone grafts strikes me as odd given that this is a highly expressed gene in islets. Is it typical for grafted islets to lose this expression? Could loss of E-cadherin be related to HIF-1 α activation? Please provide further evidence/information. Also, double check that the image contrast settings are adjusted appropriately such that important low-brightness details from raw images are not excluded from the final processed images.

Response to Reviewers

We thank reviewers for their constructive and helpful comments. Several additional experiments were performed in order to address these points and the manuscript was revised according to reviewers' queries. We believe that the revisions have significantly strengthened our manuscript.

Please note that, while answering all reviewers' queries, figures 2 and 3 were merged into a new figure 2; figure 4-6 were renumbered as figures 3-5; and a new figure 6 was added. It should also be noted that, in order to address reviewers' queries, some references were added and the reference list had to be re-numbered.

Below is a point-by-point answer to all reviewers' comments:

Reviewer #1

“This is an excellent study, reporting a cell-based strategy to improve islet cell engraftment and function for the treatment of diabetes mellitus. The strategy is based on the combination of islet cells with amniotic epithelial cells. In vitro studies showed that this combination led to a marked improvement in the viability in hypoxic conditions and in the insulin-secretory function of islet beta cells. In vivo studies were performed with rat islet cells and human amniotic epithelial cells (hAECs), transplanted in immunodeficient SCID mice rendered diabetic with streptozotocin. In this animal model, the combination of islet cells with amniotic epithelial cells determined a dramatic and long-term improvement of glycemia. Neither islet cells alone nor amniotic epithelial cells alone yielded such benefit. The glucose lowering effects were observed with the transplantation of extremely small amounts of islet cells, pre-aggregated and co-transplanted at a 1:1 ratio with amniotic epithelial cells. In vivo effects were attributed to an increase in beta cell engraftment, an increase in graft vascularization, and a restoration of cell-to matrix contact in the graft.

These claims are novel and could have a significant impact in the field of cell-based therapies for diabetes mellitus.

The in vivo results are striking: the authors report diabetes reversal in the vast majority of the recipients within 10 days after transplantation with an extremely low number of islet cells (150 organoids of islet cells and hAECs -- each composed by approximately 250 islet cells and 250 amniotic epithelial cells, for a total of approximately 37,500 islet cells and 37,500 amniotic epithelial cells per animal).

An amount of islet cells that was double than that (150 islet cell organoids, each composed by 500 islet cells, for a total of approximately 75,000 islet cells) did not determine glycemia normalization in the vast majority of the recipients for almost two months.”

We thank the reviewer for this assessment, and his/her recognition of innovative approaches and interesting findings in our work as well as for the useful recommendations for improving the manuscript.

1. There is one point that deserves to be further investigated in future studies: what is the fate of the transplanted human AECs, when co-transplanted with islets? The persistence of human AECs in the grafts could be further investigated, as well as their eventual maturation into insulin-producing beta-like cells. The investigators utilized an anti-human nuclear antigen antibody for the initial experiments of in vitro aggregation

(see Figure 2D), but did not report results with such antibody on the explanted grafts. Moreover, the *in vivo* release of human insulin/C-peptide, which can be differentiated from rat and mouse insulin/C-peptide via human-specific antibodies, could be monitored.

In response to the reviewer's major concern, which was also raised by Reviewer #2, we have now included additional findings on co-localization of hAECs within the graft as suggested. Anti-human nuclear antigen staining (supplementary Fig. 4) demonstrated that while human-derived cells were abundantly present during the first weeks, their number was gradually declining over time. At the end of a one-month period, only occasional HNA positive cells were detectable. We were unable to detect any human specific insulin and C-peptide in the plasma derived from these animals.

Interestingly, some HNA positive cells were also positive for insulin (see figure below, not included in the manuscript). Although a very striking and intriguing finding, studying the ability of hAECs to transdifferentiate into insulin-producing cells would require complex fundamental studies that are beyond the scope of the present study.

Minor comments:

2. **Results/Characterization of hAECs: HLA-G and HLA-E belong to the non-classical HLA-class Ib family.**

This information was included as recommended (page 4).

3. **Results/Generation of spheroids and assessment of *in vitro* viability and functionality:** Here and in Figure 2F the authors report the Stimulation Index (SI) [i.e. the ratio of insulin amount quantified after exposure to high glucose concentration, divided by the amount quantified after exposure to low glucose concentration]. Also, the absolute amounts of insulin would have been informative, especially if normalized by DNA content, and could have strengthened the results.

To clarify this point, we would like to specify that insulin concentration in supernatants was normalized to total insulin content of spheroid lysates as indicated in Material and Method section, page 18. We have now added the absolute insulin values and included this info in the Supplementary Table 1.

4. Results/In vivo functional assessment of transplanted islet organoids: Please indicate the n of mice that underwent IPGTT, for each group of animals.

The number of animals was indicated as requested (pages 6-7)

5. Results/Transplantation of islet organoids preserves endocrine cell mass and enhances graft revascularization: Here and in Figure 5 the authors indicate that CD34 immunostaining was used to analyze vascular endothelial cells, whereas in the Methods section (Immunohistological analyses of transplanted islet organoids) they indicate a CD31 antibody and not a CD34 antibody. Please clarify.

We have in fact utilized both CD31 and CD34 antibody staining to analyze endothelial cells. The information on the CD34 and CD31 antibodies used was added in the Methods section (page 20). CD31 immunohistology is shown as Supplementary Figures 2 and 3.

6. Results/Underlying mechanisms for improved graft function and revascularization: “but not through differentiation into the epithelial cells” ...do the author refer to endothelial cells?

We thank the reviewer for having noticed and reported this typo, which has now been corrected (page 8).

7. Discussion: State “Mesenchymal Stem (or stromal) Cells” before the acronym “MSC”.

We have now spelled out the MSC acronym (mesenchymal stem cells) in the discussion (page 11).

8. Materials and Methods: To enable replication of the experiments, please indicate the Catalog Numbers of the products used.

This was done as recommended. Catalog numbers of the products used are listed in Supplementary Tables 2-5. This is indicated at the beginning of the Methods section, page 13.

9. Materials and Methods/Isolation of human amniotic epithelial cells (hAECs): Please indicate the freezing medium utilized for cryopreservation, and indicate whether or not cryopreserved cells were used in in vivo experiments.

The freezing medium used for cryopreservation was 90% FBS and 10% DMSO. Only cryopreserved hAECs were used in the experiments. This was clarified in the Methods section (page 15).

10. Materials and Methods/Characterization of hAECs: The details of Oct-4 and HLA-E antibodies are missing.

The details of these antibodies have been added in the Methods section (page 16).

11. Materials and Methods/Cell viability and functional assessment of IC-hAEC organoids: How many clusters were handpicked?

To assess viability, at least 100 organoids per condition were analyzed. This information was added in the Materials and Methods section (page 18)

12. Materials and Methods/Diabetes induction and xenogenic islet transplantation: Please indicate here the n of spheroids handpicked.

The number of spheroids handpicked for transplantation was 150. This information was added to the Methods section (page 19)

13. Figure 5 and its Figure Legend: Please define more clearly the units of measure in the Y axes (This reviewer understands: cell number/mm² in B; % of graft area in C, and % of beta cell area in D). Please maintain consistency between the annotations in the Figures and those in the Figure Legend.

We agree that there was some confusion in the presentation of data in Figure 5. The reviewer's understanding was correct and we have clarified the units of the Y axes by making them consistent with the figure legends. Please note that Figure 5 was renumbered as Figure 4.

14. Figure 6D: The results presented in Figure 6D and 6E don't fit together. Are the columns of Figure 6D mis-labeled? Also, please make the whole Figure 6 consistent, e.g. markers in columns - and cell types in rows.

The figure was indeed mislabeled. We apologize for the confusion generated. Mislabeling was corrected and the figure was revised as per the reviewer's request. Please note that Figure 6 was renumbered as Figure 5.

Reviewer #2 (Remarks to the Author):

“The manuscript by Lebreton and colleagues describes an innovative approach to pancreatic islet transplantation whereby engrafted beta cell survival, function, and vascularization are augmented via the incorporation of human amniotic epithelial cells into re-aggregated pseudo-islet spheroids. The work is of sound technical quality. The use of hAEC as a cell type to improve grafted islet function is novel; however, it is not clear how the end results are demonstrably different or improved over other similar co-culture approaches using, for example, MSCs (e.g. DOI’s: 10.2337/db16-1068, 10.1371/journal.pone.0073526, 10.1007/s00125-011-2053-4, 10.1007/s00125-016-4120-3). The results of this study are indeed convincing for the most part, and would be interesting to the community of scientists who study islet transplant re-vascularization. However, the scope of the implications of this study for human islet transplantation are somewhat limited by the choice of the in vivo model. The study focuses mostly on transplantation outcomes, so there is only a peripheral emphasis on mechanisms (HIF-1, etc). The impact of the paper could be improved by providing a stronger translational approach that goes beyond the present results generated with rat islets transplanted into STZ-induced SCID mice.”

We thank the reviewer his/her overall positive appraisal of our work. We agree that improvement of islet engraftment and vascularization still represents the main challenge in the field, and subsequently is the main focus of current strategies in beta cell replacement. However, while pursuing the same goal, our approach fundamentally differs from the ones cited above. First, we are using epithelial cells derived from human amnion, while other groups have used mesenchymal cells, derived from different sources (mostly adipose or bone-marrow). Second, we are generating islet organoids composed of hAECs and dissociated islet cells and not shielding the islets with stem cells as the above-mentioned groups have reported. This approach is novel and allows us to reach a double goal: first, to control the size of organoids and thus to decrease islet loss caused by size-related hypoxic injury; second, and most importantly, to decrease the transplanted islet cell mass necessary by decreasing cell loss thanks to the cytoprotective effects of hAECs. Hence, the work described in this study addresses current challenges of clinical islet transplantation, and presents the first demonstration of improved viability and function of hybrid islet organoids generated from human amniotic stem cells and dissociated islet cells.

1. Please provide evidence of the fate of amniotic epithelial cells post-transplantation. How long do they survive in the islet graft? (Figure 6A is suggestive that there do not appear to be any cells other than beta cells). Or, do the AECs continue to proliferate in vivo and is there a danger of tumorigenesis in immune compromised or immune suppressed individuals?

We understand the reviewer’s concern, which was also raised by Reviewer #1. Please see answer to Reviewer#1, point 1, above.

2. To increase the broader impact of the work, it would be critical to demonstrate the efficacy of hAEC-islet organoids using primary human islets or renewable sources of human beta cells. This is because there are significant species differences between rodent and human islets in terms of vasculature, tolerance to hypoxia, and long-term graft survival.

3. The kidney capsule is a high-performing site for rodent islet transplantation but has limited potential use for human clinical islet transplantation. This is due to the limited subcapsular space that may be insufficient for placing enough human islets to control glycemia, the potential for pre-existing kidney disease in severely diabetic patients, and the risk of inducing damage to kidneys from the graft. The omentum and subcutaneous spaces have more impact for human translatability. The broader importance of the hAEC approach could be better understood in a more translatable graft location.

We fully understand these concerns and have addressed them by performing several new experiments. We have addressed both issues of applicability to human islet cells and validity of the kidney capsule implantation site in the same set of in vivo experiments. hAEC-islet organoids were generated using human islet cells and transplanted into the epididymal fat pad of murine recipients. Essentially identical results were obtained with human and rodent islet organoids, showing improved in vivo function, and revascularization. The results of these additional experiments are shown in a new paragraph in the Results section (pages 9-10; new figure 6).

4. The authors suggest that hAEC may have immunosuppressive properties leading to islet graft immune acceptance without systemic immunosuppression. Data testing this hypothesis would significantly elevate the impact of this study. Lacking such data, the demonstrated improvements to vascularization and insulin secretion alone are less broadly impactful as these metrics have been addressed before through many different strategies.

We alluded to the added immunomodulatory potential of hAECs in the Introduction and Discussion sections of our manuscript. We fully agree with the reviewer about the interest of performing our studies in an allogeneic model to study the immunomodulatory properties of hAECs and their impact on graft acceptance. As mentioned in the Discussion, we have initiated such studies in our lab with results that should be available in the coming year(s). This is however a major undertaking that falls beyond the scope of the current manuscript. On the other hand, we respectfully disagree with the reviewer on the relative lack of impact of the improvements in graft vascularization and in vivo function we have achieved in the present study.

Minor:

5. Figure 3A-F appears to be lacking an important control: the non-hypoxic condition for IC and IC+hAEC. Please include these key data for comparison.

We agree with this suggestion. Figure 3 a-f was modified to include this key information. Because of figure renumbering, for the reasons mentioned above, the data appears on new Figure 2 f-k.

6. Figure 4: From the resolution of the provided images, it is not totally clear that the CD34+ staining marks actual blood vessels and is not, rather, the hAEC cells. This ambiguity could be addressed by showing staining with a second marker such as CD31, vWF, or tomato lectin perfusion (can show vessel patency). Supplementary data should also be included to validate both antigen and species specificity of the anti-human and anti-rodent CD34 antibodies if conclusions are based solely on this marker.

We have included additional data to address these concerns. Human amniotic epithelial cells are negative for the CD31, CD34 and CD45 hematopoietic markers. We have now included FACS analysis data on these markers in Figure 1 b.

In order to show more convincing evidence, we have revised figure 4 a to show larger magnification images of CD34 staining. We have also included CD31 staining in Supplementary Figure 2.

7. Data representation: please show individual data points throughout rather than bar charts, especially for experiments with small sample sizes.

We have modified all figures to which this query applies (Figures 2-6) according to reviewer's recommendation.

8. I counted two instances of the phrase "data not shown." Rather than excluding these data, please include it as supplementary information.

We fully understand this concern and have added the data in the Supplementary files as recommended. We have included data on spheroid viability in normoxic conditions (Supplementary Figure 1) and on negative staining of human specific CD31 demonstrating that vascularization of the graft arose from the host but not from hAECs (Supplementary Figure 3).

9. Figure 6G: The total lack of E-cadherin staining in islet-alone grafts strikes me as odd given that this is a highly expressed gene in islets. Is it typical for grafted islets to lose this expression? Could loss of E-cadherin be related to HIF-1 α activation? Please provide further evidence/information. Also, double check that the image contrast settings are adjusted appropriately such that important low-brightness details from raw images are not excluded from the final processed images.

We thank the reviewer for raising this interesting question. As suspected by the reviewer, the impression that islet-alone grafts have a total lack of E-cadherin expression is due to low brightness in comparison to a renal tubule shown in the picture. Figure 5 h shows that E-cadherin, albeit at a much lower level than in full organoids, is indeed expressed. We have changed Figure 5 g to better show the difference in staining brightness between renal tubules and IC-spheroids.

To the best of our knowledge, no publication has studied E-cadherin expression in islets of Langerhans after transplantation, and we can only have speculative answers to how typical is the loss of E-cadherin expression in grafted islets. Several groups have shown that a decrease in E-cadherin expression takes place under hypoxic conditions in other cell types, mostly in tumor cell lines, but also in an intestinal cell line, which is consistent with our observation. We have slightly expanded the discussion on this point and added 2 new references (pages 12-14; new references 37 and 38).

Decreased E-cadherin expression and HIF-1 α activation can appear concurrently, but the mechanistic link is unclear (Imai et al. doi: 10.1016/S0002-9440(10)63501-8 ; Zhu et al. doi: 10.1002/path.5089). However, such a mechanism is not involved in our findings, since the effects of hypoxia on islet-alone spheroids were to decrease both E-cadherin (Supplementary Figure 6) and HIF-1 α expression (Figure 2 i-j) in comparison with islet-hAEC organoids.

Reviewers' comments:

Reviewer #1 (Remarks to the Author):

This is an excellent study reporting the successful generation of composite micro-organs comprised of islet cells and human amniotic epithelial cells. Incorporation of the amniotic cells into the islet organoids

not only improved their viability and function, but also resulted in better engraftment and improved post-transplant function, even when a marginal organoid mass was implanted. These results indicate that composite amniotic and islet cell organoids could be of assistance to improve cell based therapies for diabetes. The methods are appropriate and the results justify the conclusions of this excellent study.

Camillo Ricordi, MD, FNAI

Reviewer #2 (Remarks to the Author):

The authors have done a good job of addressing many of the concerns raised by reviewers 1 and 2. The manuscript has been much improved by the inclusion of the new human islet cell transplant experiment to the epididymal fat pad.

It is important to be clear about what this advancement does and does not do. The hAEC IC technology strongly addresses major gaps and challenges in islet transplantation of insufficient islet mass, poor vascularization, and hypoxia. However, the current manuscript does not address the major issues of replenishable alternative beta cell sources or lifelong immunosuppression that greatly restrict the availability of current human pancreas/islet transplantation. In light of this point, I still have the following remaining major concern:

1. I still wonder about my comment #4 suggesting further study of the hypothesized immunosuppressive properties of hAECs. The authors agree that this point is important, but claim that this topic is best suited for a future study due to the extensive time and effort it would take to investigate. I do agree that this issue is not trivially addressed, but at the same I argue that it is not out of reach because there are many potential in vivo or in vitro experiments that could be performed that would provide some insight. I should clarify that I did not actually mean to suggest an allograft is the only way to investigate this topic, only that the issue of immunomodulation should be dealt with in some manner at the discretion of the authors. Information of the hypothesized immune suppressive properties of hAECs is important to establish for this technology to be touted as a viable strategy for allo/xenograft alone. Alternatively, it should be established that the current data on incorporation of hAECs only addresses vascularization and must be combined with other strategies for immune modulation for clinical translation.

This topic is not only a matter of this reviewer's opinion but also reflects the direction of the conversation within the field and the stated policy of major funding agencies (e.g. JDRF) that invest in islet replacement technologies. My point is, if the authors want to claim that their model "has great potential in the development of cell-based therapies for type 1 diabetes" while suggesting that the hAECs have immunosuppressive properties, I challenge them to substantiate this claim with primary data on the potential for immunomodulation. Otherwise, the authors need to be clear in their claims and conclusions about what hAECs can and cannot do in their model.

In addition to the above point, I note the following remaining minor issues:

2. In the Figure 1b flow cytometry results, it isn't clear what n=16 means. Is this the number of technical replicates or experimental replicates using different hAEC donors? Please be specific here as to the meaning of n.

3. Scale bars are not defined in figure captions for some images (e.g. Figure 2a, 3d, 3g, S6).
4. The meaning of error bars is stated for some figures but not in others. Please use a consistent formatting for figure captions throughout.
5. The caption for Figure 6 does not indicate the site of transplantation (epididymal fat pad). It is important to be explicit here as readers may miss this key detail.
6. Likewise the caption for Supplementary Figure 5 does not indicate the species of IC used or transplant site and contains a typo, "orgaboids."
7. The islet spheroids are described as "fully functional" at three places in the manuscript (abstract, introduction, discussion) but the only functional assessments performed were static GSIS in vitro and an in vivo glycemic control. Certainly, these measurements indicate that the spheroids are responsive to glucose stimulation by secreting insulin. However, there are other important factors to consider when declaring an islet organoid to be "fully" functional. Other measures of islet function might include but are not limited to appropriate glucagon secretion, first and second phase glucose responsive insulin kinetics during perfusion, and islet synchronization and interconnectedness assessed by Ca²⁺ signaling. The authors should reconsider their usage of the word "fully" as it might be more accurate to simply state that the spheroids are "functional" or "glucose-responsive."
8. Not related to the content of the manuscript, but as a comment to the authors, I suggest they consider their new finding that some rare, HNA-positive cells are also insulin-positive (included in the response letter but not the manuscript) with an abundance of caution. To substantiate such a claim, it would be critical to include a cell membrane marker to clearly delineate the boundaries of different cells and provide very clear high-resolution images. For instance, it could be the case that the depicted hAEC might simply be surrounded by or on top of a larger beta cell in this 2D section of a more complex 3D structure, creating the illusion of insulin-positivity. Lineage tracing studies would very likely be required to prove this. I think there should be a very high burden of proof required to make this claim.

Point-by-point response to Reviewers

Reviewer #1 (Remarks to the Author):

This is an excellent study reporting the successful generation of composite micro-organs comprised of islet cells and human amniotic epithelial cells. Incorporation of the amniotic cells into the islet organoids not only improved their viability and function. But also resulted in better engraftment and improved post-transplant function, even when a marginal organoid mass was implanted. These results indicate that composite amniotic and islet cell organoids could be of assistance to improve cell based therapies for diabetes. The methods are appropriate and the results justify the conclusions of this excellent study.

Camillo Ricordi, MD, FNAI

We thank Dr. Ricordi for his very positive comments.

Reviewer #2 (Remarks to the Author):

The authors have done a good job of addressing many of the concerns raised by reviewers 1 and 2. The manuscript has been much improved by the inclusion of the new human islet cell transplant experiment to the epididymal fat pad.

It is important to be clear about what this advancement does and does not do. The hAEC IC technology strongly addresses major gaps and challenges in islet transplantation of insufficient islet mass, poor vascularization, and hypoxia. However, the current manuscript does not address the major issues of replenishable alternative beta cell sources or lifelong immunosuppression that greatly restrict the availability of current human pancreas/islet transplantation. In light of this point, I still have the following remaining major concern:

1. I still wonder about my comment #4 suggesting further study of the hypothesized immunosuppressive properties of hAECs. The authors agree that this point is important, but claim that this topic is best suited for a future study due to the extensive time and effort it would take to investigate. I do agree that this issue is not trivially addressed, but at the same I argue that it is not out of reach because there are many potential in vivo or in vitro experiments that could be performed that would provide some insight. I should clarify that I did not actually mean to suggest an allograft is the only way to investigate this topic, only that the issue of immunomodulation should be dealt with in some manner at the discretion of the authors. Information of the hypothesized immune suppressive properties of hAECs is important to establish for this technology to be touted as a viable strategy for allo/xenograft alone. Alternatively, it should be established that the current data on incorporation of hAECs only addresses vascularization and must be combined with other strategies for immune modulation for clinical translation.

This topic is not only a matter of this reviewer's opinion but also reflects the direction of the conversation within the field and the stated policy of major funding agencies (e.g. JDRF) that invest in islet replacement technologies. My point is, if the authors want to claim that their model "has great potential in the development of cell-based therapies for type 1 diabetes" while suggesting that the hAECs have immunosuppressive properties, I challenge them to substantiate this claim with primary data on the potential for immunomodulation. Otherwise, the authors need to be clear in their claims and conclusions about what hAECs can and cannot do in their model.

We agree with most of the issues raised by the Reviewer. However, we would like to argue that they illustrate the complexity of the situation and that our study does indeed provide a strategy that has the potential to address at least some of these issues.

Successful long-term beta-cell replacement, as exemplified by islet of Langerhans transplantation, is currently hampered by two major issues: 1) poor engraftment, mostly as a result of inflammation at the implantation site and hypoxia due to impaired revascularization, and 2) the need for lifelong immunosuppression with the associated toxicity profiles.

It is true that for a strategy to be translatable to the human, the immunosuppression issue must also be addressed, and that this is within the policy of the major funding agency investing in the field (JDRF). It is however fair to note that in JDRF "roadmap" (<http://grantcenter.jdrf.org/wp-content/uploads/2019/07/Beta-Cell-Replacement-Program-Strategy.pdf>), they acknowledge that "*the first generation therapeutic product will likely consist of a renewable cell dose in an open scaffold protected by standard, broad immune suppression. The development of next generation and aspirational products will focus on strategies to deliver insulin producing cells from a renewable source without broad immune suppression*", in other words, they foresee a 2-stage development, in which immune modulatory strategies would only occur in the second stage. We believe, and we thank the Reviewer for agreeing with this point, that we have indeed addressed quite successfully the engraftment issue, and that this is a significant achievement.

We also agree that the immunomodulatory issue must be tackled next, and this is what we wrote at the end of the discussion of our paper: "We are currently investigating in immunocompetent animals whether the immunomodulatory properties conferred by hAECs would allow maintenance of the grafts with minimal or no immunosuppression." At this point, we have never claimed to have shown any degree of immune protection conferred by hAECs to our constructs. However, the immunomodulatory properties of hAECs have been studied and reported in the literature (for example: doi: 10.1111/aji.13003, doi: 10.2147/SCCAA.S58696).

To summarize the major concern raised by Reviewer #2, he "challenges (us) to substantiate this claim with primary data on the potential for immunomodulation. Otherwise, the authors need to be clear in their claims and conclusions about what hAECs can and cannot do in their model." We propose to reply to this query in the following way:

1. We have modified the last paragraph as follows: "Other authors have demonstrated the immunomodulatory properties conferred by amniotic cells (new references 42,43), an observation that we have also made with our hAECs in preliminary experiments. We will next investigate whether these immunomodulatory properties could allow maintenance of the grafts with minimal or no immunosuppression".
2. As mentioned above, we have generated primary data demonstrating the immunomodulatory potential of our hAECs. In two sets of experiments, we have shown that third party hAECs were able to abrogate immune response in a dose-dependent manner in MLRs. We have added these data in the supplementary file (Supplementary Figure 7).
3. As a consequence of the addition of these 2 new references, previous references 42-46 have been renumbered 44-48.

In addition to the above point, I note the following remaining minor issues:

2. In the Figure 1b flow cytometry results, it isn't clear what n=16 means. Is this the number of technical replicates or experimental replicates using different hAEC donors? Please be specific here as to the meaning of n.

We agree that this needed to be clarified. The data are expressed as mean \pm SEM in cell preparations obtained from 16 different donors. We have now clarified this in the legend to Figure 1b.

3. Scale bars are not defined in figure captions for some images (e.g. Figure 2a, 3d, 3g, S6).

Missing scale bars from these figures have been defined in their corresponding legends.

4. The meaning of error bars is stated for some figures but not in others. Please use a consistent formatting for figure captions throughout.

As indicated in the Methods section, all results are presented as mean \pm SEM. To make the figure captions consistent, we have added this indication to the legends of Figures 2-6.

5. The caption for Figure 6 does not indicate the site of transplantation (epididymal fat pad). It is important to be explicit here as readers may miss this key detail.

We would like to point out that the site of transplantation is clearly indicated in the title to the figure caption: "In vivo function of a human derived organoids transplanted in the epididymal fat pad."

6. Likewise the caption for Supplementary Figure 5 does not indicate the species of IC used or transplant site and contains a typo, "orgaboids."

We thank the reviewer for having noticing this. The typo has been corrected, the species of IC (human) has been indicated, as well as the transplant site (epididymal fat pad).

7. The islet spheroids are described as "fully functional" at three places in the manuscript (abstract, introduction, discussion) but the only functional assessments performed were static GSIS in vitro and an in vivo glycemic control. Certainly, these measurements indicate that the spheroids are responsive to glucose stimulation by secreting insulin. However, there are other important factors to consider when declaring an islet organoid to be "fully" functional. Other measures of islet function might include but are not limited to appropriate glucagon secretion, first and second phase glucose responsive insulin kinetics during perfusion, and islet synchronization and interconnectedness assessed by Ca²⁺ signaling. The authors should reconsider their usage of the word "fully" as it might be more accurate to simply state that the spheroids are "functional" or "glucose-responsive."

We have considered this query by omitting the word "fully" in each instance in which islet spheroid functionality was described (page 3, line 5; page 4, line 23; page 11, line 13).

8. Not related to the content of the manuscript, but as a comment to the authors, I suggest they consider their new finding that some rare, HNA-positive cells are also insulin-positive (included in the response letter but not the manuscript) with an abundance of caution. To substantiate such a claim, it would be critical to include a cell membrane marker to clearly delineate the boundaries of different cells and provide very clear high-resolution images. For instance, it could be the case that the depicted hAEC might simply be surrounded by or on top of a larger beta cell in this 2D section of a more complex 3D structure, creating the illusion of insulin-positivity. Lineage tracing studies would very likely be required to prove this. I think there should be a very high burden of proof required to make this claim.

We thank the reviewer for these useful suggestions on how this observation could be further studied. This comment relates to data and a figure shown to the referees in our response to their first review, but not included in the manuscript.

REVIEWERS' COMMENTS:

Reviewer #2 (Remarks to the Author):

The revised manuscript is of sound technical quality and the authors have satisfactorily responded to the peer review comments by conducting additional experiments that strengthen their conclusions. I have no further recommendations.